# ELIGN: Expectation Alignment as a Multi-Agent Intrinsic Reward

**Zixian Ma**[1]**, Rose E. Wang**[1]**, Li Fei-Fei**[1]**, Michael Bernstein**[1]**, Ranjay Krishna**[1,2]

Stanford University[1], University of Washington[2]

{zixianma,rewang,feifeili,msb,ranjaykrishna}@cs.stanford.edu

## Abstract

Modern multi-agent reinforcement learning frameworks rely on centralized training and reward shaping to perform well. However, centralized training and dense rewards are not readily available in the real world. Current multi-agent algorithms struggle to learn in the alternative setup of decentralized training or sparse rewards. To address these issues, we propose a self-supervised intrinsic reward *ELIGN - expectation alignment -* inspired by the self-organization principle in Zoology. Similar to how animals collaborate in a decentralized manner with those in their vicinity, agents trained with expectation alignment learn behaviors that match their neighbors' expectations. This allows the agents to learn collaborative behaviors without any external reward or centralized training. We demonstrate the efficacy of our approach across 6 tasks in the multi-agent particle and the complex Google Research football environments, comparing ELIGN to sparse and curiosity-based intrinsic rewards. When the number of agents increases, ELIGN scales well in all multi-agent tasks except for one where agents have different capabilities. We show that agent coordination improves through expectation alignment because agents learn to divide tasks amongst themselves, break coordination symmetries, and confuse adversaries. These results identify tasks where expectation alignment is a more useful strategy than curiosity-driven exploration for multi-agent coordination, enabling agents to do zero-shot coordination.

## 1   Introduction

Many real world AI applications can be formulated as multi-agent systems, including autonomous vehicles (Cao et al., 2012), resource management (Ying & Dayong, 2005), traffic control (Sunehag et al., 2017), robot swarms (Swamy et al., 2020), and multi-player video games (Berner et al., 2019). Agents must adapt their behaviors to each other in order to coordinate successfully in these systems. However, adaptive coordination algorithms are challenging to develop because each agent is not privy to other agents' intentions and their future behaviors (Foerster et al., 2017).

These challenges are more acute in decentralized training under partial observability than centralized training or full observability. In the real world, agents act under partial observability and learn in a decentralized manner: they do not learn collaborative behaviors with a single centralized algorithm with a complete knowledge of the environment (Iqbal & Sha, 2019; Liu et al., 2020). Unfortunately, the most successful multi-agent algorithms train agents with a centralized critic, assuming access to all agents' observations and actions (Foerster et al., 2018; Rashid et al., 2018; Sunehag et al., 2017; Lowe et al., 2017). The most successful multi-agent algorithms for decentralized training and partial observability assume task-specific reward shaping (Jain et al., 2020; Iqbal & Sha, 2019), which is expensive to generate. These algorithms struggle to learn with sparse reward structure.

Consider a cooperative navigation task, where $N$ agents aim to simultaneously occupy $N$ goal locations. A centralized algorithm with full observability is capable of optimally assigning the nearest

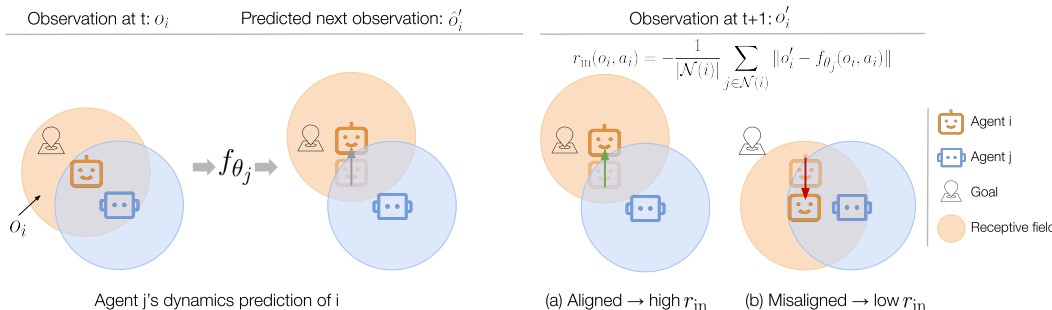

Figure 1: We introduce ELIGN, i.e, expectation alignment, a task-agnostic intrinsic reward to improve multi-agent systems. Intuitively, ELIGN encourages agents to become more predictable to their neighbors. An agent (e.g., agent $i$ here) learns to behave in ways that match its neighbors' (e.g., agent $j$'s) predictions of its next observation. Here, agent $j$ expects agent $i$ to move up instead of down, moving closer to a point of interest above it. Agent $i$ attains (a) a higher reward when its action (e.g., upward) aligns with this expectation or (b) a lower reward when its action (e.g., downward) is misaligned.

goal location to each respective agent. However, with partial observability, agents can see only a handful of goal locations and other agents. They are unaware of others' observations, actions, and intentions with decentralized training. We observe that agents simultaneously occupy the same goal; they fail to collaborate because they do not predict which goal each agent is expected to occupy. To overcome instances of miscoordination, decentralized algorithms have adapted single-agent curiosity-based intrinsic rewards (Pathak et al., 2017; Stadie et al., 2015). Multi-agent curiosity-based rewards incentivize agents to explore novel states (Iqbal & Sha, 2020). Although curiosity helps agents discover new goal locations, it doesn't solve the challenge of coordination, such as assigning goals to each agent. Only a few attempts explore other forms of multi-agent intrinsic rewards (Iqbal & Sha, 2020; Böhmer et al., 2019; Schafer, 2019).

In this work, we propose ELIGN as a novel multi-agent self-supervised intrinsic reward, enabling decentralized training under partial observability. Intuitively, expectation alignment encourages agents to elicit behaviors that decrease future uncertainty for their team: it encourages each agent to choose actions that match their teammates' expectations. Going back to the cooperative navigation task, expectation alignment encourages each agent to move to goals others expect it to occupy, like goals that are either closest to the agent or goals that other agents aren't moving towards (Figure 1). We take inspiration from the self-organization principle in Zoology (Couzin, 2007). This principle hypothesizes that collective animal intelligence emerges because groups synchronize their behaviors using only their local environment; they do not rely on complete information about other agents and can coordinate successfully by predicting the dynamics of agents within their field-of-view (Collett et al., 1998; Theraulaz & Bonabeau, 1995; Ben-Jacob et al., 1994; Buhl et al., 2006). Similarly, expectation alignment as an intrinsic reward is calculated based on the agent's local observations and its approximation of neighboring agents' expectations. It does not require a centralized controller nor full observability. ELIGN is task-agnostic and we apply it to both collaborative and competitive multi-agent tasks.

We demonstrate the efficacy of our approach in the multi-agent particle and Google Research football environments, two popular benchmarks for multi-agent reinforcement learning (Lowe et al., 2017; Kurach et al., 2019). We evaluate ELIGN under partial and full observability, with decentralized and centralized training, and in terms of scalability. We observe that expectation alignment outperforms sparse and curiosity-based intrinsic rewards (Ndousse et al., 2021; Stadie et al., 2015; Iqbal & Sha, 2020), especially under partial observability with decentralized training. We additionally test expectation alignment as a way to perform zero-shot coordination with new agent partners, and investigate why ELIGN improves coordination. We show that agent coordination improves through expectation alignment because agents learn to divide tasks amongst themselves and break coordination symmetries (Hu et al., 2020).

## 2 Related Work

Our formulation of expectation alignment, a task-agnostic intrinsic reward for multi-agent training, draws inspiration from the self-organization principle in Zoology, which posits that synchronized group behavior is mediated by local behavioral rules (Couzin, 2007) and not by a centralized controller (Camazine et al., 2020). Group cohesion emerges by predicting and adjusting one's behavior to that of near neighbors (Buhl et al., 2006). This principle underlies the coordination found in multi-cellular organisms (Camazine et al., 2020), the migration of wingless locusts (Collett et al., 1998), the collective swarms of bacteria (Ben-Jacob et al., 1994), the construction of bridge structures by ants (Theraulaz & Bonabeau, 1995), and some human navigation behaviors (Couzin, 2007).

**Intrinsic motivation for single agents.** Although we draw inspiration from Zoology for formalizing expectation alignment as an intrinsic reward, there is a rich body of work on intrinsic rewards within the single-agent reinforcement learning community. To incentivize exploration, even when non-optimal successful trajectories are uncovered first, scholars have argued for the use of intrinsic motivation (Schmidhuber, 1991). Single-agent intrinsic motivation has focused on exploring previously unencountered states (Pathak et al., 2017; Burda et al., 2018a), which works particularly well in discrete domains. In continuous domains, identifying unseen states requires keeping track of an intractable number of visited states; instead, literature has recommended learning a forward dynamics model to predict future states and identify novel states using the uncertainty of this model (Achiam & Sastry, 2017). Other formulations encourage re-visiting states where the dynamics model's prediction of future states errs (Stadie et al., 2015; Pathak et al., 2017). Follow up papers have improved how uncertainty (Kim et al., 2020) and model errors (Burda et al., 2018b; Sekar et al., 2020) are calculated.

**Intrinsic motivation for multiple agents.** Most multi-agent intrinsic rewards have been adapted from single-agent curiosity-based incentives (**?**Böhmer et al., 2019; Schafer, 2019) and have primarily focused on cooperative tasks. They propose intrinsic rewards to improve either coordination, collaboration, or deception: These rewards either maximize information conveyed by an agent's actions (**?**Chitnis et al., 2020; Wang et al., 2019), shape the influence of an agent (Jaques et al., 2019; Foerster et al., 2017), incentivize agents to hide intentions (Strouse et al., 2018), build accurate models of other agents' policies (Hernandez-Leal et al., 2019; Jaques et al., 2019), or break extrinsic rewards for better credit assignment (Du et al., 2019).

Several multi-agent intrinsic rewards (Hernandez-Leal et al., 2019; Jaques et al., 2019), including ours, rely on the ability to model others' dynamics in a shared environment. This ability is a key component to coordination, closely related to Theory of Mind (Tomasello et al., 2005). Our work can be interpreted as using a Theory of Mind model of others' behaviors to calculate an intrinsic motivation loss. Unlike existing Theory of Mind methods that learn a model per collaborator (Roy et al., 2020), we learn a single dynamics model, allowing our method to scale as the number of agents increase. Our proposal is related to model-based reinforcement learning (Jaderberg et al., 2016; Wang et al., 2020a); however, instead of learning a dynamics model for control, we learn a dynamics model as a source of reward. Our work is closely related to a recently proposed auxiliary loss on predicting an agent's own future states (Ndousse et al., 2021). However, there are three key differences. First, their work predicts ego-agent observations, whereas our work additionally predicts future observations from the other agents' point of view. Second, their loss optimizes state embeddings while ours optimizes agents' policies. Third, their work focuses on cooperative tasks whereas ours applies to both cooperative and competitive domains.

**Multi-agent reinforcement learning algorithms.** Today, the predominant deep multi-agent framework uses actor-critic methods with a centralized critic and decentralized execution (Lowe et al., 2017; Foerster et al., 2018; Iqbal & Sha, 2019; Liu et al., 2020; Rashid et al., 2018). This framework allows a critic to access the observations and actions of all agents to ease training. However, there are several situations where centralized training may not be desirable or possible. Examples include low bandwidth communication restrictions or human-robot tasks where observations cannot be easily shared between agents (Ying & Dayong, 2005; Cao et al., 2012; Huang et al., 2015). Decentralized training is therefore the most practical training paradigm but it suffers from unstable training: the environment is nonstationary from a single-agent's perspective (Lowe et al., 2017). Our work uses a decentralized training framework and tackles the nonstationarity challenge with an intrinsic reward designed to improve an agent's ability to model others. We also apply expectation alignment to centralized training and observe that it still aids cooperative and some competitive tasks.

## 3 Background

We formulate our setting as a partially observable Markov game $(\mathcal{S}, \mathcal{O}, \mathcal{A}, \mathcal{T}, r_{\text{ex}}, N)$ (Littman, 1994). A Markov game for $N$ agents is defined by a state space $\mathcal{S}$ describing the possible configurations of the environment. The observation space for agents is $\mathcal{O} = (\mathcal{O}_1, \ldots, \mathcal{O}_N)$ and the action space is $\mathcal{A} = (\mathcal{A}_1, \ldots, \mathcal{A}_N)$. Each agent $i$ observes $\mathbf{o}_i \in \mathcal{O}_i$, a private partial view of the state, and performs actions $a_i \in \mathcal{A}_i$. Using the observation, each agent uses a stochastic policy $\pi_{\theta_i} : \mathcal{O}_i \times \mathcal{A}_i \to [0, 1]$, where $\theta_i$ parameterizes the policy. The environment changes according to the state transition function which transitions to the next state using the current state and each agent's actions, $\mathcal{T} : \mathcal{S} \times \mathcal{A} \to \mathcal{S}$. The team of agents obtains a shared extrinsic reward as a function of the environment state, $r_{\text{ex}} : \mathcal{S} \times \mathcal{A} \to \mathbb{R}$. The team's goal is to maximize the total expected return: $R = \sum_{t=0}^{T} \gamma^t r_{\text{ex}}^t$ where $0 \leq \gamma \leq 1$ is the discount factor, $t$ is the time step, and $T$ is the time horizon. The environment may also contain adversarial agents who have their own reward structure.

## 4 Expectation Alignment

To understand expectation alignment intuitively, let's revisit the cooperative navigation task, where $N$ agents are rewarded for simultaneously occupying as many goal locations as possible. In Figure 1, agent $i$ has a dynamics model trained on its past experiences. It predicts how future states will evolve from the point of view of agent $j$, who is within $i$'s view. In this example, $j$ will expect $i$ to move towards the goal since $i$ is closer to it. ELIGN encourages $i$ to pursue the action that $j$ expects (Figure 1(a)). In turn, $j$ can now assume that the observed goal location will eventually be occupied by $i$ and should therefore explore to find another goal. By aligning shared expectations, agent behaviors become more predictable. Conversely, when neighbors behave opposite to an agent's predictions, the agent can infer about the environment outside of its own receptive field (Krause et al., 2002). For example, in Figure 1 (b), if agent $j$ observes $i$ running away from a goal, this surprising behavior might indicate the existence of an adversary outside $j$'s receptive field.

Our training algorithm consists of three interwoven phases of learning a dynamics model, calculating the ELIGN reward, and optimizing the agent's policy (Algorithm 1).

### 4.1 Training the dynamics model

Similar to prior work (Wang et al., 2018; Kidambi et al., 2020), each agent $i$ learns a dynamics model $f_{\theta_i}$ to predict the next observation $\hat{o}'_i$ given its current observation and action $o_i, a_i$, i.e,

$$\hat{o}'_i = f_{\theta_i}(o_i, a_i).$$

We use a three-layer Multi-Layer Perceptron with ReLU non-linearities as the dynamics model. We minimize the mean squared error between its prediction and ground truth next observation $o'_i$.

### 4.2 Calculating intrinsic reward

The intrinsic reward captures how well agent $i$ aligns to its neighbors' (e.g., agent $j$'s) expectations on its next state. Calculating this reward requires $j$ to accurately predict $i$'s behavior, simulating a Theory of Mind (Tomasello et al., 2005). As suggested by the self-organization principle, $i$ must learn to align to $j$'s predictions. Ideally, the ELIGN intrinsic reward is calculated as:

$$r_{\text{in}}(o_i, a_i) = -\frac{1}{|\mathcal{N}(i)|} \sum_{j \in \mathcal{N}(i)} \|o'_i - f_{\theta_j}(o_i, a_i)\|$$

where $\mathcal{N}(i)$ is the set of neighbors within $i$'s receptive field, including $i$ itself. The ELIGN reward is high when the average $L_2$ loss is small, i.e, when $i$'s actual next observation is close to agent $j$'s predicted observation of $i$ for all $j$ in its neighbors. In that case, $i$ has chosen an action that aligns with $j$'s expectations of how $i$ should act.

In a decentralized training setup, however, $i$ doesn't have access to $j$'s dynamics model $f_{\theta_j}$, so $i$ approximates $j$'s dynamic model with a proxy: its own dynamics model $f_{\theta_i}$ and the knowledge of agent $j$'s observation radius. Such an approximation is ecologically valid since we often approximate others' behaviors using a second-order cognitive Theory of Mind (Morin, 2006). Additionally, $i$

doesn't have access to $j$'s entire observation; so, we restrict the future prediction from $j$'s point of view by using the portion of $j$'s observation $i$ can see: $o_{i \cap j} = o_i \odot o_j$. Agent $i$'s decentralized intrinsic reward then becomes:

$$r_{\text{in}}(o_i, a_i) = -\frac{1}{|\mathcal{N}(i)|} \sum_{j \in \mathcal{N}(i)} \|o'_{i \cap j} - f_{\theta_i}(o_{i \cap j}, a_i)\|$$

We found that the approximation of $f_{\theta_j}$ using $f_{\theta_i}$ works well empirically. Dynamics model losses for all agents quickly decrease within 5-10 training epochs. we validate its applicability in small-scale heterogeneous multi-agent tasks where agents have variable capabilities, although we find the methods perform similarly when more heterogeneous agents are added.

### 4.3 Policy learning

Once the ELIGN rewards are calculated, the total rewards at each step for each agent $i$ is: $r_i = r_{\text{ex}} + \beta r_{\text{in}}(o_i, a_i)$ where $r_{\text{ex}}$ is the extrinsic reward provided by the environment and $\beta$ is a hyper-parameter for weighing the intrinsic reward in the agent's overall reward calculation. In practice, we set $\beta$ to be $\frac{1}{|\mathcal{O}_i|}$ where $|\mathcal{O}_i|$ is the observation dimension; we find this scale generalizes well across tasks. Since our contribution is agnostic to any particular multi-agent training algorithm, the team of agents can now be trained using any multi-agent training algorithm to maximize returns $R = \sum_{t=0}^{T} \gamma^t r$.

Both centralized and decentralized training algorithms can make use of these rewards. We primarily use the multi-agent decentralized variant of the soft-actor critic algorithm in our experiments (Haarnoja et al., 2018; Iqbal & Sha, 2019). Compared to centralized joint-action training, whose action space grows exponentially in $N$ agents, our decentralized method has linear space complexity. Further, decentralized training can parallelize training time to be less than linear with respect to $N$. Although we present results with one centralized training framework, studying the impact of expectation alignment with all the centralized critic frameworks is out of scope for this paper.

---

**Algorithm 1** ELIGN: Expectation Alignment

1: Initialize replay buffer $D$ and $D'$
2: Initialize $N$ agents with random $\theta_i$: $i \in [1, N]$
3: **while** not converged **do**
4:    **for** $b = 1 \ldots B$ **do**
5:       Populate buffer $D$ with episode using policies $(\pi_{\theta_1}, \ldots, \pi_{\theta_N})$
6:    **end for**
7:    // TRAIN DYNAMICS MODEL
8:    **for** agent $i = 1 \ldots N$ **do**
9:       Sample transitions: $\{(o_i, a_i, r_{\text{ex}}, o'_i)\} \sim D_i$
10:      Predict $\hat{o}'_i = f_{\theta_i}(o_i, a_i)$
11:      Update dynamics $\theta_i$ using $o'_i$.
12:    **end for**
13:    // CALCULATE ELIGN REWARD
14:    **for** agent $i = 1 \ldots N$ **do**
15:      Sample $B$ transitions: $\{(o_i, a_i, r_{\text{ex}}, o'_i)\} \sim D_i$
16:      Compute intrinsic rewards $r_{\text{in}}(o_i, a_i)$
17:      Add $\{(o_i, a_i, r_{\text{ex}} + \beta r_{\text{in}}, o'_i)\}$ to $D'_i$
18:    **end for**
19:    // POLICY LEARNING
20:    Update all $\theta_i$s using transitions from $D'$
21: **end while**

---

### 4.4 Extending expectation alignment to competitive tasks

We extend the ELIGN formulation to competitive tasks where a team of agents compete against adversaries. In this case, agents are encouraged to *misalign* with their adversaries' expectations, i.e, agents are incentivized to be unpredictable to their adversaries within its receptive field ($\mathcal{N}_{\text{adv}}(i)$):

$$r_{\text{in}} = \frac{1}{|\mathcal{N}_{\text{adv}}(i)|} \sum_{k \in \mathcal{N}_{\text{adv}}(i)} \|o'_{i \cap k} - f_{\theta_i}(o_{i \cap k}, a_i))\|$$

## 5 Experiments

Our experiments explore the utility of using expectation alignment as an intrinsic reward compared to sparse and curiosity-based intrinsic rewards. We primarily focus on decentralized training under partial observability. However, we also demonstrate that ELIGN can easily augment centralized methods and assist in fully observable tasks. We vary the number of agents in the multi-agent particle tasks to test scalability. We end by investigating how and why ELIGN improves coordination by designing three evaluation conditions. First, does expectation alignment improve coordination by

Table 1: We report the mean test episode extrinsic rewards and standard errors of *decentralized training under partial observability* in multi-agent particle and Google Research football environments. ELIGN$_{\text{self/team}}$ outperform SPARSE and both curiosity-based intrinsic rewards. ELIGN$_{\text{adv}}$ achieves the best performance among all competitive tasks except for *Physical deception*, where ELIGN$_{\text{team}}$ is the best. These results demonstrate the benefit of using alignment as intrinsic reward to train better decentralized policies under partial observability.

| | Cooperative | | Competitive | | | |
|---|---|---|---|---|---|---|
| Task (Agt# v Adv#) | Coop nav. (3v0) | Hetero nav. (4v0) | Phy decep. (2v1) | Pred-prey (2v2) | Keep-away (2v2) | 3v1 w/ keeper (3v2) |
| SPARSE[1] | $139.07 \pm 13.63$ | $284.42 \pm 12.83$ | $93.60 \pm 8.61$ | $-4.72 \pm 2.4$ | $4.58 \pm 3.27$ | $0.020 \pm 0.001$ |
| CURIO$_{\text{self}}^2$ | $133.93 \pm 7.66$ | $286.22 \pm 9.97$ | $68.80 \pm 7.93$ | $-6.50 \pm 2.18$ | $11.88 \pm 2.88$ | $0.024 \pm 0.004$ |
| CURIO$_{\text{team}}^3$ | $125.42 \pm 11.95$ | $262.28 \pm 22.59$ | $85.31 \pm 11.93$ | $-3.57 \pm 1.75$ | $9.54 \pm 5.04$ | $0.021 \pm 0.002$ |
| ELIGN$_{\text{self}}$ | $\mathbf{155.88 \pm 5.11}$ | $292.34 \pm 9.24$ | $69.91 \pm 4.51$ | $-7.58 \pm 2.55$ | $12.84 \pm 4.29$ | $0.003 \pm 0.018$ |
| ELIGN$_{\text{team}}$ | $141.04 \pm 8.04$ | $\mathbf{311.67 \pm 10.88}$ | $\mathbf{101.72 \pm 6.31}$ | $-7.69 \pm 2.69$ | $2.96 \pm 4.03$ | $0.022 \pm 0.001$ |
| ELIGN$_{\text{adv}}$ | — | — | $92.20 \pm 4.23$ | $\mathbf{-2.51 \pm 1.70}$ | $\mathbf{19.46 \pm 5.05}$ | $\mathbf{0.025 \pm 0.001}$ |
| HAND-CRAFTED[1] | $75.56 \pm 18.90$ | $228.48 \pm 18.88$ | $94.25 \pm 14.75$ | $-0.77 \pm 0.17$ | $52.14 \pm 3.11$ | — |

[1] Lowe et al. (2017); Kurach et al. (2019),[2] Stadie et al. (2015),[3] Iqbal & Sha (2020)

Test Occupancy (↑)/Collision (↓) Count Per Step

Figure 2: We plot the average test occupancy/collision count per step of decentralized algorithms under partially observable multi-agent particle tasks. On these metrics, ELIGN$_{\text{self}}$ and ELIGN$_{\text{adv}}$ perform the best on cooperative and competitive tasks respectively.

breaking symmetries (Hu et al., 2020; Wang et al., 2020b)? Second, does ELIGN enable zero-shot generalization to new partners? Lastly, is the dynamics model critical in aligning agent behaviors?

## 5.1 Environments

We evaluate ELIGN across both cooperative and competitive tasks in the multi-agent particle environment (Mordatch & Abbeel, 2017; Lowe et al., 2017) and the Google Research football environment (Kurach et al., 2019).

**State and action space** The multi-agent particle environment is a two-dimensional world. The Google Research football environment is a three-dimensional world. Both environments have continuous state spaces and discrete action spaces. Particle agents observe all agents' positions and velocities. They can "stay" or change their velocity in one of the four cardinal directions. Each football agent controls one player. Players observe the ball, other players' positions and directions. They can apply one of ten actions from "top_left", "top", "top_right", "right", "bottom_right", "bottom", "bottom_left", "sprint", and "dribble".

**Observability** The original environments assume full observability, where each agent can observe the position $p = (x, y)$ and velocity $v = (\Delta x, \Delta y)$ of all agents; each agent's observation vector is thus $\mathbf{o}_{i,\text{full}} = [p_1, \ldots, p_N, v_1, \ldots, v_N]$. We extend these environments to be partially observable, where agent $i$ observes only the portion within its receptive field; like prior work with partial observability (Corder et al., 2019), we hide the position and velocity information of any agent $j$ outside of agent $i$'s receptive field; i.e, if the Euclidean distance between agent $i$ and $j$ surpasses a vicinity threshold $\tau$, then $p_j$ and $v_j$ are 0 in $\mathbf{o}_{i,\text{partial}}$. We set $\tau = 0.5$ for partially observable and $\infty$ in the original fully observable case, where the world's width and height are 2.0 in the multi-agent particle environment and $0.84 : 2.00$ in the Google Research football environment. Both environments also contain objects such as obstacles, goals and a ball; they are similarly hidden if out of sight. Partial observability is a more ecologically valid training condition since most agents in real-world tasks can only observe a small portion of their environment at a given time.

## 5.2 Tasks

**Multi-agent particle environment** We use the following tasks from the multi-agent particle environment (Lowe et al., 2017; Liu et al., 2020). We choose $N$ based on prior work.

Table 2: We report the mean test episode extrinsic rewards and standard errors as the number of agents is increased and trained using of *decentralized* algorithms. When the number of agents increases, one of ELIGN still performs the best in all tasks except for *Heterogenous navigation*.

| Task (Agt # vs. Adv #) | Cooperative | | Competitive | | |
|---|---|---|---|---|---|
| | Coop nav. (5v0) | Hetero nav. (6v0) | Phy decep. (4v2) | Pred-prey (4v4) | Keep-away (4v4) |
| SPARSE[1] | $459.92 \pm 22.44$ | $616.62 \pm 25.30$ | $166.89 \pm 27.72$ | $-28.75 \pm 7.3$ | $0.75 \pm 1.82$ |
| CURIO$_{self}^2$ | $458.45 \pm 19.79$ | $\mathbf{702.73 \pm 18.57}$ | $146.55 \pm 29.05$ | $-25.35 \pm 6.16$ | $10.52 \pm 5.48$ |
| CURIO$_{team}^3$ | $497.15 \pm 11.47$ | $695.38 \pm 12.22$ | $84.66 \pm 16.94$ | $-17.21 \pm 8.23$ | $1.40 \pm 2.06$ |
| ELIGN$_{self}$ | $\mathbf{498.24 \pm 9.77}$ | $646.70 \pm 23.25$ | $137.38 \pm 30.00$ | $\mathbf{-9.14 \pm 5.57}$ | $9.83 \pm 11.22$ |
| ELIGN$_{team}$ | $488.83 \pm 20.82$ | $638.74 \pm 28.93$ | $\mathbf{186.83 \pm 21.92}$ | $-20.4 \pm 5.93$ | $2.07 \pm 4.55$ |
| ELIGN$_{adv}$ | — | — | $182.61 \pm 17.63$ | $-21.37 \pm 7.02$ | $\mathbf{11.29 \pm 9.02}$ |

[1] Lowe et al. (2017),[2] Stadie et al. (2015),[3] Iqbal & Sha (2020)

*Cooperative navigation:* $N$ agents must cooperate to reach a set of $N$ goal locations. Agents are collectively rewarded based on the occupancy of any agent on any goal location.

*Heterogeneous navigation:* $N$ agents must reach $N$ goals but they differ in speeds and sizes. $\frac{N}{2}$ agents are slow and big, and the other $\frac{N}{2}$ agents are fast and small.

*Physical deception:* $N$ agents cooperate to reach a single goal location and are rewarded if any one occupies the goal. However, they are penalized when any of $\frac{N}{2}$ adversaries occupies the goal and gets rewarded. The adversaries do not know which landmark is the goal and must infer it from the agents' behavior. The agents should learn to deceive the adversaries by covering all the landmarks.

*Keep-away:* There are $N$ landmarks, one of which is the goal and known to $N$ agents. Agents are rewarded for occupying it and preventing $M$ adversaries from reaching it. Adversaries are rewarded for pushing the agents away from the goal, but they can only infer which landmark is the goal.

*Predator-prey:* $N$ slow adversaries chase and capture $N$ fast cooperating agents around a randomly generated obstacle-filled environment. Each time an adversary catches an agent, the agent is penalized and the adversary is rewarded.

**Google Research football** We use the *Academy 3vs1 with Keeper* competitive task from the Google Research football environment (Kurach et al., 2019). In this task, three agents try to score from the edge of the penalty box, one on each side, and the other at the center. This task is initialized with the centered agent having the ball and facing the defender. There is an adversary who plays the keeper.

## 5.3 Training and evaluation

We train all algorithms with 5 random seeds. All the hyperparameters used in the training can be found in the Appendix. For the Multi-agent particle environment, each experiment uses one Tesla K40 GPU to train until convergence, i.e. the best evaluation episode reward hasn't changed for 100 epochs. Each epoch equates to $4K$ episodes of 25 timesteps. We evaluate the algorithms by running $1K$ test episodes of 25 timesteps and mainly report the mean average test episode reward and standard error across the random seeds. We also evaluate on task-specific metrics, including agent-goal occupancy/agent-adversary collision count, and agent-goal/agent-adversary distance. For Google Research football, each experiment trains for $5M$ timesteps. We evaluate on and report the mean average episode rewards and the standard errors across the seeds.

## 5.4 Baselines

All algorithms are trained using the same agent architecture and optimization algorithm. They vary in task-specific extrinsic rewards and intrinsic rewards. We use two versions of the soft actor-critic algorithm Haarnoja et al. (2018): a decentralized one that trains each agent individually without access to other agents' observations and actions (ie. the original soft-actor critic algorithm) and a centralized one with a critic that has access to other agents' observations and actions (Iqbal & Sha, 2019). Note, our intrinsic reward can also be added to non-actor-critic methods, such as COMA (Foerster et al., 2018) and VDN (Sunehag et al., 2017). We leave this to future work to avoid conflating the effects of expectation alignment with COMA's counterfactual reasoning and VDN's value decomposition.

We use SPARSE (Lowe et al., 2017; Kurach et al., 2019), CURIO$_{self}$ (Stadie et al., 2015), CURIO$_{team}$ (Iqbal & Sha, 2020), and variations of our ELIGN rewards. SPARSE rewards agents

only when they reach a goal state. CURIO$_{\text{team}}$ is a curiosity-based multi-agent intrinsic reward which maximizes the average $L_2$ loss (instead of minimizing it in ELIGN). It rewards agents for exploring novel states (Iqbal & Sha, 2020). CURIO$_{\text{self}}$ also maximizes the $L_2$ loss but only using agent $i$'s own observation (Stadie et al., 2015). We experiment with three variants of ELIGN: ELIGN$_{\text{self}}$, incentivizing alignment to one's own expectation ELIGN$_{\text{team}}$ incentivizing agents to align to their team, and ELIGN$_{\text{adv}}$ incentivizing misalignment to adversaries' expectations. Note that ELIGN$_{\text{self}}$ is similar to the auxiliary loss in Ndousse et al. (2021) but we use it for policy optimization, rather than for training a state encoder. We also add hand-crafted dense rewards to provide oracle performance for all tasks.

## 5.5 Results in partially observable environments with decentralized training

**ELIGN outperforms baselines across cooperative and competitive tasks in the multi-agent particle environment.** Table 1 demonstrates that both ELIGN$_{\text{self}}$ and ELIGN$_{\text{team}}$ outperform all SPARSE and CURIO$_{\text{self/team}}$ baseline rewards in cooperative tasks. While not all ELIGN variants surpass the baselines in competitive tasks, we find that ELIGN$_{\text{team}}$ achieves the highest reward in *Physical deception*, and ELIGN$_{\text{adv}}$ performs the best in *Predator-prey* and *Keep-away*. Similarly, Figure 2 shows that ELIGN$_{\text{self}}$ achieves the highest per-step occupancy count in both cooperative tasks, and ELIGN$_{\text{adv}}$ does the best in all competitive tasks.

**ELIGN outperforms baselines in the complex Google Research football environment.** As shown in Table 1, ELIGN$_{\text{adv}}$ achieves the best mean average episode reward in the competitive *Academy 3vs1 with keeper* task. Collectively, these results provide empirical evidence that the self-organizing principle improves coordination under partial information, a setting that is most realistic to real world multi-agent systems.

**In competitive tasks, agents benefit more from being misaligned to adversaries than being aligned to their team members.** Among the four competitive tasks in the multi-agent particle and football environments, we find that ELIGN$_{\text{adv}}$ outperforms all SPARSE and CURIO$_{\text{self/team}}$ baselines and other variants of ELIGN in *Predator-prey*, *Keep-away* and *Academy 3vs1 with keeper*. This suggests that being misaligned to adversaries, ie taking surprising actions that conflict with the adversary's expectations, might be a more useful strategy in multi-agent competitive tasks.

**When the number of agents increases, ELIGN scales well in all multi-agent particle tasks except for *Heterogenous navigation*.** Table 2 shows that our ELIGN intrinsic reward still largely achieves the best performance when more agents are added to cooperative and competitive tasks. The only exception is the *Heterogenous navigation* task, where both ELIGN$_{\text{self}}$ and ELIGN$_{\text{team}}$ outperform SPARSE but not CURIO$_{\text{self/team}}$. We hypothesize that it is more difficult for agents to predict their neighbors' future states accurately when there are more agents with different sizes and speeds, and errors in dynamics prediction could lead to misleading alignment signals. Further, we see a consistent increase in ELIGN$_{\text{team}}$'s performance compared against SPARSE even when the number of agents scales to ten in *Cooperative navigation* (Figure 3).

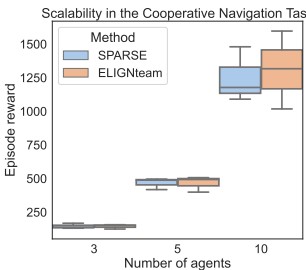

Figure 3: ELIGN$_{\text{team}}$ achieves consistent gains compared against SPARSE when the number of agents increases in the *Cooperative navigation* task.

## 5.6 Results with full observability and centralized ELIGN

We further test the utility of decentralized ELIGN in fully observable environments and centralized ELIGN under partial observability. We find that decentralized expectation alignment helps in fully observable *Cooperative navigation*, *Heterogenous navigation*, and *Predator-prey*, tasks where expectation alignment has been observed in nature. Similarly, centralized ELIGN also improves agents' performance compared against SPARSE and CURIO$_{\text{self/team}}$ rewards in the same tasks with partial observability. These results can be found in Tables 3 and 4 in the Appendix. As full observability and centralized training are our main focus, we leave it to future work to investigate why expectation alignment benefits these tasks but not others.

## 5.7 Investigating how the ELIGN reward helps

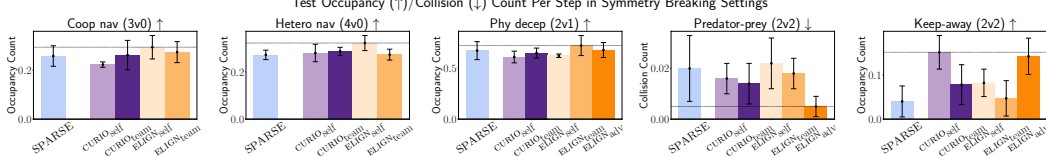

Figure 5: We plot the test occupancy/collision count per step of decentralized algorithms in symmetry-breaking settings under partial observability. ELIGN$_{self}$ performs the best in both cooperative tasks. ELIGN$_{team}$ and ELIGN$_{adv}$ are the best strategies in *Physical deception* and *Predator-prey*.

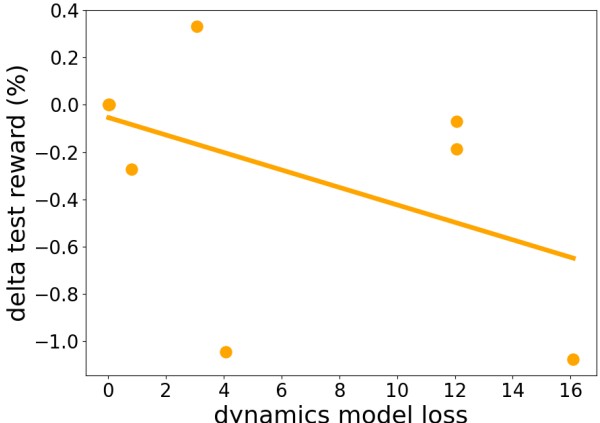

Figure 6: Test performance decreases with dynamics model loss ($R^2 = 0.242$), implying that ELIGN requires an accurate dynamics model.

We further investigate how expectation alignment improves coordination through three evaluation setups.

**ELIGN helps agents divide subtasks.** A core challenge in multi-agent collaboration is efficient task division (Wang et al., 2020b). Here, we test whether expectation alignment improves sub-task allocation. We initialize agents in states without an optimal

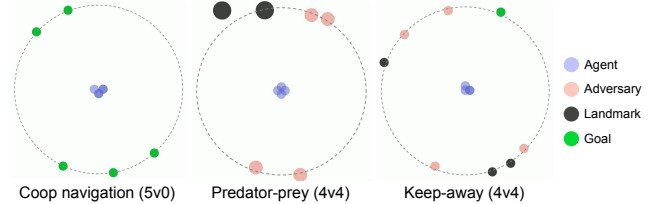

Figure 4: We visualize the symmetry-breaking setups in three example tasks. More details can be found in the Appendix.

sub-task allocation, necessitating symmetry-breaking (Hu et al., 2020). Figure 4 illustrates the symmetry-breaking setups: in cooperative navigation, when agents are initialized equidistant to all the goal locations, there isn't an optimal allocation of agents to goals. We find that ELIGN$_{self}$ achieves the best performance in both cooperative tasks, while ELIGN$_{team}$ and ELIGN$_{adv}$ are the best strategies in *Physical deception* and *Predator-prey* respectively (Figure 5). Upon a qualitative evaluation of *Cooperative navigation*, we observe that agents with expectation alignment are able to predict which goals will be covered by their collaborators and move towards their allocated one. Without expectation alignment, agents move towards the same goal.

**ELIGN helps agents generalize to new partners.** Another core challenge in multi-agent collaboration is zero-shot coordination, where agents are tested to collaborate with new partners they haven't been trained with. We study whether expectation alignment enables better zero-shot coordination. New partners are sampled from other training runs with different seeds and the team is evaluated using the same metrics as before. We observe that ELIGN strategies enable better performance than SPARSE on average, and one of ELIGN$_{self,team,adv}$ performs the best in *Heterogenous navigation*, *Physical deception* and *Keep away*. (Figure 7). These results suggest that ELIGN results in better zero-shot coordination with new partners sampled from separate training runs.

**Accuracy of the dynamics model affects ELIGN.** We investigate the accuracy of the dynamics model in calculating useful intrinsic rewards. Since ELIGN uses a dynamics model to calculate

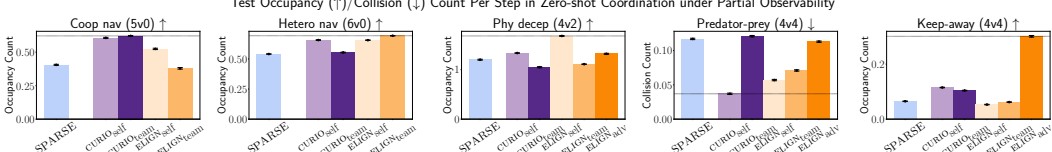

Figure 7: We plot the test occupancy/collision count per step of decentralized algorithms in zero-shot evaluation with new partners. We find that one of ELIGN$_{\text{self,team,adv}}$ achieves the best performance in *Heterogenous navigation*, *Physical deception* and *Keep away*.

rewards, we test whether an inaccurate model misleads agents towards unaligned behaviors. We train agents on noisy dynamics models by adding Gaussian noise $\epsilon \sim \mathcal{N}(0, \sigma)$ to the predictions made by the dynamics model. We run experiments with multiple $\sigma$ values to study how performance changes as the dynamics model becomes more noisy: $\sigma \in [0.5, 1.0, 2.0]$. Our experiments cover one cooperative and one competitive task. Figure 6 plots the *final* dynamics loss against the reward change from the noiseless run. As the dynamics model degrades, we observe that the task performance also drops. This study identifies the importance of an accurate dynamics model, suggesting that expectation alignment should used in environments where an accurate dynamics model can be learned.

## 6    Discussion

**Limitations and future work.** While curiosity has proven useful for exploration in single-agent tasks, we find that expectation alignment—which mathematically encourages agents to be more predictable instead of finding novelty—outperforms curiosity in numerous multi-agent tasks. We hypothesize that our results arise because today's multi-agent task state space requires significantly less exploration than those used for single-agent (e.g. Atari games). Our findings are limited to the multi-agent particle and Google Research football environments, which have a smaller action space than most ecologically valid scenarios.

Language, motion, and human gesture are all combinatorially vast; in such action spaces, expectation alignment might develop social dynamics that hinder non-optimal multi-agent behaviors. Similarly, photorealistic environments have a larger state space, where teams perform common household activities (e.g., cooking, cleaning, etc.) or drive together in crowded cities (Srivastava et al., 2021). Future work should develop new multi-agent environments that demand exploration complexity and where both curiosity and expectation alignment would be necessary for collaboration. For example, in a search and rescue task where a single agent is unable to carry the injured, curiosity would encourage "search" while ELIGN would speed up "rescue". In the end, we envision that both these forms of rewards would be necessary for successful collaboration. However, choosing when to encourage curiosity versus expectation alignment is an open research problem.

Additionally, enabling stable multi-agent training without centralized training could open up future opportunities for legible (Dragan et al., 2013) agents in human environments. Agents with interpretable actions can induce more faithful human mental models, improving human-AI interaction; however, predictability does not imply legibility. Future work could explore the role of legibility in designing intrinsic rewards.

Future work should also explore the use of expectation alignment in massive collaboration settings with hundreds of agents. Drawing on Zoology research Couzin (2007) expectation alignment should scale in such settings if agents align their behaviors only to their nearest neighbors and not to the entire cohort.

**Conclusion.** Inspired by the self-organizing principle in Zoology, we introduce ELIGN, i.e, expectation alignment, a simple, task-agnostic, and self-supervised intrinsic reward for multi-agent systems. ELIGN rewards agents when they act predictably to their teammates and unpredictably to their adversaries. ELIGN improves multi-agent performance across six cooperative and competitive tasks in the multi-agent particle and Google Research football environments, especially for decentralized training under partial observability. It also scales well, helps agents break symmetries, and generalize to new partners.

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
