# ELIGN: Expectation Alignment
# as a Multi-Agent Intrinsic Reward

**Zixian Ma**[1], **Rose E. Wang**[1], **Li Fei-Fei**[1], **Michael Bernstein**[1], **Ranjay Krishna**[1,2]

Stanford University[1], University of Washington[2]

{zixianma,rewang,feifeili,msb,ranjaykrishna}@cs.stanford.edu

## Abstract

Modern multi-agent reinforcement learning frameworks rely on centralized training and reward shaping to perform well. However, centralized training and dense rewards are not readily available in the real world. Current multi-agent algorithms struggle to learn in the alternative setup of decentralized training or sparse rewards. To address these issues, we propose a self-supervised intrinsic reward *ELIGN - expectation alignment -* inspired by the self-organization principle in Zoology. Similar to how animals collaborate in a decentralized manner with those in their vicinity, agents trained with expectation alignment learn behaviors that match their neighbors' expectations. This allows the agents to learn collaborative behaviors without any external reward or centralized training. We demonstrate the efficacy of our approach across 6 tasks in the multi-agent particle and the complex Google Research football environments, comparing ELIGN to sparse and curiosity-based intrinsic rewards. When the number of agents increases, ELIGN scales well in all multi-agent tasks except for one where agents have different capabilities. We show that agent coordination improves through expectation alignment because agents learn to divide tasks amongst themselves, break coordination symmetries, and confuse adversaries. These results identify tasks where expectation alignment is a more useful strategy than curiosity-driven exploration for multi-agent coordination, enabling agents to do zero-shot coordination.

## References

Iqbal, S. and Sha, F. Coordinated exploration via intrinsic rewards for multi-agent reinforcement learning, 2020. URL https://openreview.net/forum?id=rkltEOVKwH.

Kurach, K., Raichuk, A., Stanczyk, P., Zajac, M., Bachem, O., Espeholt, L., Riquelme, C., Vincent, D., Michalski, M., Bousquet, O., and Gelly, S. Google research football: A novel reinforcement learning environment. *CoRR*, abs/1907.11180, 2019. URL http://arxiv.org/abs/1907.11180.

Liang, E., Liaw, R., Moritz, P., Nishihara, R., Fox, R., Goldberg, K., Gonzalez, J. E., Jordan, M. I., and Stoica, I. URL https://arxiv.org/abs/1712.09381.

Lowe, R., Wu, Y., Tamar, A., Harb, J., Abbeel, P., and Mordatch, I. Multi-agent actor-critic for mixed cooperative-competitive environments. *Advances in Neural Information Processing Systems*, 2017.

Stadie, B. C., Levine, S., and Abbeel, P. Incentivizing exploration in reinforcement learning with deep predictive models. *arXiv preprint arXiv:1507.00814*, 2015.

Weng, J., Chen, H., Yan, D., You, K., Duburcq, A., Zhang, M., Su, H., and Zhu, J. Tianshou: A highly modularized deep reinforcement learning library. *arXiv preprint arXiv:2107.14171*, 2021.

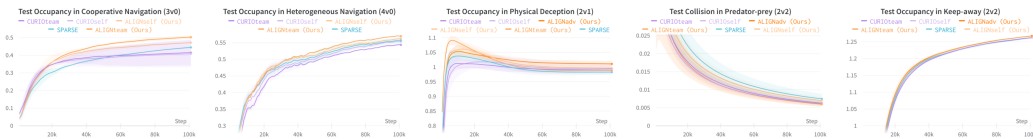

Figure 1: Learning curves of test occupancy/collision in all five tasks in the multi-agent particle environment. On average, it takes the best ELIGN variant 65 epochs to reach the maximum score of the best CURIO method at 100 epochs, which means ELIGN requires on average 35% fewer samples than CURIO.

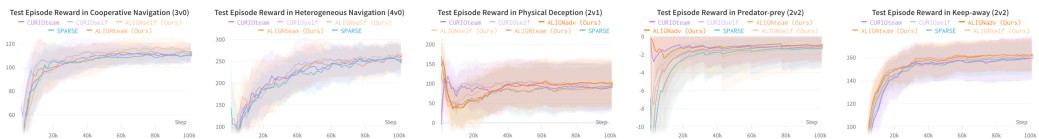

Figure 2: Learning curves of test episode rewards in all five tasks in the multi-agent particle environment.

## A  Appendix

### A.1  Code

We upload our code for training and evaluating agents with and without expectation alignment in both the multi-agent particle and Google Research football environments here: `https://github.com/StanfordVL/alignment`.

### A.2  Symmetry-breaking initializations

We create a symmetry-breaking version of each task for evaluation by initializing the environment in the following ways:

*Cooperative Navigation and Heterogenous Navigation:* All agents are initialized at the origin (i.e. center of the world), and target landmarks are placed randomly on a circle perimeter with the maximum radius (i.e. world radius - the greatest landmark size) so that each agent is equidistant from each target landmark.

*Physical Deception:* Both agents and adversaries start at the origin. All the landmarks, including the goal, are randomly initialized on a circle perimeter.

*Predator-prey:* The collaborative agents are initialized at the center while the adversaries are placed randomly on a circle perimeter. All the landmarks are randomly initialized in the world.

*Keep-away:* All the cooperative agents are placed at the origin. Adversaries and landmarks, including the goal, are randomly initialized on a circle perimeter. In this task setup, we do not initialize the adversaries at the center because they are awarded for colliding with the cooperative agents.

### A.3  Learning curves

Compared to the best baseline's highest performance at the 100th epoch, we find that it only takes the best ELIGN variant 31 (Coop nav), 74 (Hetero nav), 30 (Physical dec), 97 (Predator-prey) and 92 (Keep-away) epochs respectively in the five multi-agent particle tasks (1). This means that on average the best ELIGN variant requires $35 \pm 32$ fewer training steps to reach the same performance as curiosity or spare methods.

## A.4 Assets and licenses

We use four assets in total, two of which are existing multi-agent reinforcement learning environments, and the other two are libraries for training reinforcement learning algorithms.

We conduct our evaluation on the multi-agent particle (Lowe et al., 2017) and Google Research football (Kurach et al., 2019) environments, which are under the MIT license and Apache-2.0 license respectively.

We adapt the tianshou (Weng et al., 2021) and rllib (part of the ray package) (Liang et al.) libraries to our experiments, and they are under the MIT license and Apache-2.0 license respectively.

## A.5 Societal impacts

While developing new intrinsic rewards to improve decentralized multi-agent training can help develop and deploy agents in a variety of applications, we foresee no immediate societal consequences of this work. However, our experiments thus far have not studied the possible degradation of behaviors when agents align to malicious teammates. We have also not tested how emergent properties promote better or worse human collaborators.

## A.6 Additional tables

We include 20 tables of additional results that quantify the agents' performance under fully observable environments, with centralized training, and beyond extrinsic reward.

Table 1 reports the test episode rewards of decentralized methods under the fully observable multi-agent particle environment.

Table 2 reports the test episode rewards of centralized methods under partial observability.

Table 3 and 4 report two sets of metrics of *decentralized* methods trained with different intrinsic rewards in both partially and fully observable settings. Table 3 reports the average number of agent-target occupancies per step (or, we can understand it as: on average, the total number of goals occupied by the agents at any given timestep throughout an episode) and agent-adversary collisions in *Predator-prey*. Higher scores are better for the occupancy metric, and lower scores are better for collision. Table 4 reports the average minimum agent-to-target distance and agent-to-adversary distance. Agent-to-target distances measure the closest distance an agent achieves to the target location; lower scores are better on this metric. Agent-to-adversary distances measure the closest distance an adversary gets to a good agent; higher scores are better on this metric. Note that these distance-based metrics are not included in the reward functions, and should mainly be used to make comparisons in the case where primary metrics (i.e, reward and occupancy/collision count) have the same values.

Table 5 and 6 report the same metrics as 3 and 4 respectively, but in *scaled* environments with more agents.

Tables 7, 8 and 9 report the test episode reward and additional metrics of *decentralized* algorithms in the *symmetry-breaking* experiments conducted under "Investigating how alignment reward helps". Table 10, 11 and 12 report the same set of metrics but from experiments conducted in *scaled* and *symmetry-breaking* environments.

Table 13 reports the test mean episode rewards of *centralized* algorithms with different intrinsic rewards under full observability. Table 14 and 15 show the other two sets of metrics (i.e, occupancy/collision count and agent-target/agent-adversary distance) of *centralized* algorithms. Table 16, 17, and 18 contain the same metrics as 13, 14 and 15 respectively, but in *scaled* environments.

Finally, Tables 19 and 20 report the test episode reward values and secondary distance-based metrics for the zero-shot generalization experiments conducted under "Investigating how ELIGN reward helps". These experiments measure how well agents trained on different seeds generalized to new partners trained on other seeds.

Table 1: We report the mean test episode extrinsic rewards and standard errors of *decentralized* methods with different intrinsic rewards in fully observable environments.

| Task (Agt # vs. Adv #) | | Cooperative | | Competitive | | |
| --- | --- | --- | --- | --- | --- | --- |
| | | Coop nav. (3v0) | Hetero nav. (4v0) | Phy decep. (2v1) | Pred-prey (2v2) | Keep-away (2v2) |
| **Full observability** | SPARSE[1] | $154.00 \pm 10.51$ | $274.75 \pm 19.74$ | $82.97 \pm 12.23$ | $-10.48 \pm 4.20$ | $\mathbf{4.95 \pm 2.96}$ |
| | $\text{CURIO}_{\text{self}}$ | $154.71 \pm 8.00$ | $268.85 \pm 15.61$ | $\mathbf{100.66 \pm 15.14}$ | $-8.74 \pm 4.62$ | $-2.00 \pm 1.24$ |
| | $\text{ELIGN}_{\text{self}}$ | $\mathbf{161.70 \pm 4.52}$ | $\mathbf{280.16 \pm 17.12}$ | $87.50 \pm 15.40$ | $\mathbf{-5.60 \pm 2.60}$ | $0.40 \pm 1.92$ |

[1] Lowe et al. (2017)

Table 2: We report the mean test episode extrinsic rewards and standard errors of *centralized* methods with different intrinsic rewards under partial observability.

| Task (Agt # vs. Adv #) | | Cooperative | | Competitive | | |
| --- | --- | --- | --- | --- | --- | --- |
| | | Coop nav. (3v0) | Hetero nav. (4v0) | Phy decep. (2v1) | Pred-prey (2v2) | Keep-away (2v2) |
| **Partial observability** | SPARSE[1] | $113.25 \pm 8.10$ | $178.62 \pm 9.62$ | $\mathbf{117.45 \pm 10.63}$ | $-1.96 \pm 1.45$ | $\mathbf{35.79 \pm 14.93}$ |
| | $\text{CURIO}_{\text{self}}^{2}$ | $128.77 \pm 7.70$ | $190.30 \pm 7.73$ | $111.08 \pm 10.09$ | $-1.63 \pm 1.27$ | $13.94 \pm 12.56$ |
| | $\text{CURIO}_{\text{team}}^{3}$ | $114.13 \pm 11.84$ | $189.80 \pm 11.81$ | $114.32 \pm 5.46$ | $-3.04 \pm 1.09$ | $6.01 \pm 3.36$ |
| | $\text{ELIGN}_{\text{self}}$ | $\mathbf{137.14 \pm 3.63}$ | $169.58 \pm 14.99$ | $93.27 \pm 3.70$ | $-0.41 \pm 0.28$ | $22.77 \pm 9.91$ |
| | $\text{ELIGN}_{\text{team}}$ | $119.10 \pm 10.89$ | $\mathbf{210.81 \pm 9.70}$ | $96.49 \pm 6.46$ | $-0.92 \pm 0.72$ | $24.94 \pm 12.58$ |
| | $\text{ELIGN}_{\text{adv}}$ | — | — | $102.37 \pm 6.98$ | $\mathbf{-0.13 \pm 0.03}$ | $8.70 \pm 4.44$ |

[1] Lowe et al. (2017),[2] Stadie et al. (2015),[3] Iqbal & Sha (2020)

## A.7 Model architecture and hyperparameters

Table 21 presents the model architecture and hyperparameters used to train the algorithms in the multi-agent particle and Google Research football environments.

Table 3: The average test occupancy/collision count per step and standard errors of *decentralized* methods with different intrinsic rewards under partial and full observability. Higher scores are better for the occupancy metric (↑), and lower scores are better for the collision metric (↓).

| Task (Agt # vs. Adv #) | | Cooperative | | Competitive | | |
|---|---|---|---|---|---|---|
| | | Coop nav. (3v0) ↑ | Hetero nav. (4v0) ↑ | Phy decep. (2v1) ↑ | Pred-prey (2v2) ↓ | Keep-away (2v2) ↑ |
| **Partial observability** | SPARSE | $0.46 \pm 0.05$ | $0.57 \pm 0.01$ | $0.98 \pm 0.07$ | $0.02 \pm 0.01$ | $0.07 \pm 0.02$ |
| | $CURIO_{self}$ | $0.43 \pm 0.03$ | $0.60 \pm 0.01$ | $0.99 \pm 0.03$ | $0.02 \pm 0.01$ | $0.14 \pm 0.02$ |
| | $CURIO_{team}$ | $0.42 \pm 0.05$ | $0.59 \pm 0.01$ | $0.95 \pm 0.03$ | $0.02 \pm 0.01$ | $0.10 \pm 0.03$ |
| | $ELIGN_{self}$ | $0.52 \pm 0.03$ | $0.61 \pm 0.01$ | $0.95 \pm 0.02$ | $0.03 \pm 0.01$ | $0.10 \pm 0.02$ |
| | $ELIGN_{team}$ | $0.44 \pm 0.04$ | $0.58 \pm 0.02$ | $0.99 \pm 0.07$ | $0.03 \pm 0.01$ | $0.07 \pm 0.02$ |
| | $ELIGN_{adv}$ | — | — | $1.00 \pm 0.06$ | $0.01 \pm 0.01$ | $0.15 \pm 0.03$ |
| **Full observability** | SPARSE | $0.46 \pm 0.11$ | $0.57 \pm 0.01$ | $0.88 \pm 0.09$ | $0.03 \pm 0.01$ | $0.06 \pm 0.02$ |
| | $CURIO_{self}$ | $0.50 \pm 0.07$ | $0.59 \pm 0.02$ | $1.09 \pm 0.13$ | $0.03 \pm 0.01$ | $0.02 \pm 0.00$ |
| | $ELIGN_{self}$ | $0.48 \pm 0.11$ | $0.58 \pm 0.02$ | $0.83 \pm 0.10$ | $0.02 \pm 0.01$ | $0.04 \pm 0.01$ |

Table 4: The average test agent-to-target (agt-target) and agent-to-adversary (agt-adv) distances and standard errors of *decentralized* methods with different intrinsic rewards under partial and full observability. Lower scores are better for agt-target (↓), and higher scores are better for agt-adv (↑).

| Task (Agt # vs. Adv #) | | Cooperative | | Competitive | | |
|---|---|---|---|---|---|---|
| | | Coop nav. (3v0) ↓ | Hetero nav. (4v0) ↓ | Phy decep. (2v1) ↓ | Pred-prey (2v2) ↑ | Keep-away (2v2) ↓ |
| **Partial observability** | SPARSE | $0.30 \pm 0.02$ | $0.23 \pm 0.00$ | $0.26 \pm 0.01$ | $1.45 \pm 0.11$ | $1.41 \pm 0.07$ |
| | $CURIO_{self}$ | $0.32 \pm 0.02$ | $0.25 \pm 0.01$ | $0.25 \pm 0.00$ | $1.36 \pm 0.06$ | $1.14 \pm 0.09$ |
| | $CURIO_{team}$ | $0.31 \pm 0.01$ | $0.25 \pm 0.01$ | $0.26 \pm 0.00$ | $1.48 \pm 0.13$ | $1.31 \pm 0.10$ |
| | $ELIGN_{self}$ | $0.33 \pm 0.03$ | $0.25 \pm 0.01$ | $0.26 \pm 0.00$ | $1.39 \pm 0.12$ | $1.26 \pm 0.09$ |
| | $ELIGN_{team}$ | $0.33 \pm 0.02$ | $0.23 \pm 0.01$ | $0.25 \pm 0.01$ | $1.38 \pm 0.13$ | $1.38 \pm 0.09$ |
| | $ELIGN_{adv}$ | — | — | $0.25 \pm 0.01$ | $1.54 \pm 0.08$ | $1.14 \pm 0.09$ |
| **Full observability** | SPARSE | $0.32 \pm 0.09$ | $0.23 \pm 0.00$ | $0.26 \pm 0.01$ | $1.23 \pm 0.12$ | $1.27 \pm 0.09$ |
| | $CURIO_{self}$ | $0.28 \pm 0.04$ | $0.22 \pm 0.01$ | $0.23 \pm 0.01$ | $1.37 \pm 0.15$ | $1.53 \pm 0.03$ |
| | $ELIGN_{self}$ | $0.30 \pm 0.07$ | $0.23 \pm 0.01$ | $0.27 \pm 0.01$ | $1.40 \pm 0.13$ | $1.41 \pm 0.10$ |

Table 5: The average test occupancy/collision count per step and standard errors of *decentralized* methods with different intrinsic rewards in *scaled* environments under partial and full observability. Higher scores are better for the occupancy metric (↑), and lower scores are better for the collision metric (↓).

| Task (Agt # vs. Adv #) | | Cooperative | | Competitive | | |
|---|---|---|---|---|---|---|
| | | Coop nav. (5v0) ↑ | Hetero nav. (6v0) ↑ | Phy decep. (4v2) ↑ | Pred-prey (4v4) ↓ | Keep-away (4v4) ↑ |
| **Partial observability** | SPARSE | $0.50 \pm 0.04$ | $0.46 \pm 0.08$ | $1.20 \pm 0.10$ | $0.11 \pm 0.02$ | $0.08 \pm 0.02$ |
| | $CURIO_{self}$ | $0.48 \pm 0.03$ | $0.63 \pm 0.01$ | $1.20 \pm 0.08$ | $0.07 \pm 0.02$ | $0.15 \pm 0.06$ |
| | $CURIO_{team}$ | $0.53 \pm 0.03$ | $0.60 \pm 0.02$ | $1.20 \pm 0.09$ | $0.05 \pm 0.02$ | $0.06 \pm 0.01$ |
| | $ELIGN_{self}$ | $0.49 \pm 0.03$ | $0.67 \pm 0.00$ | $1.30 \pm 0.23$ | $0.04 \pm 0.02$ | $0.14 \pm 0.08$ |
| | $ELIGN_{team}$ | $0.56 \pm 0.04$ | $0.56 \pm 0.00$ | $1.21 \pm 0.09$ | $0.08 \pm 0.02$ | $0.10 \pm 0.02$ |
| | $ELIGN_{adv}$ | — | — | $1.23 \pm 0.10$ | $0.08 \pm 0.02$ | $0.16 \pm 0.07$ |
| **Full observability** | SPARSE | $0.52 \pm 0.11$ | $0.46 \pm 0.08$ | $0.99 \pm 0.09$ | $0.21 \pm 0.01$ | $0.03 \pm 0.00$ |
| | $CURIO_{self}$ | $0.39 \pm 0.13$ | $0.56 \pm 0.01$ | $0.86 \pm 0.04$ | $0.16 \pm 0.03$ | $0.04 \pm 0.00$ |
| | $ELIGN_{self}$ | $0.55 \pm 0.11$ | $0.56 \pm 0.00$ | $1.04 \pm 0.07$ | $0.15 \pm 0.03$ | $0.06 \pm 0.02$ |

Table 6: The average test agent-to-target (agt-target) and agent-to-adversary (agt-adv) distances and standard errors of *decentralized* methods with different intrinsic rewards in *scaled* environments under partial and full observability. Lower scores are better for agt-target (↓), and higher scores are better for agt-adv (↑).

| Task (Agt # vs. Adv #) | | Cooperative | | Competitive | | |
|---|---|---|---|---|---|---|
| | | Coop nav. (5v0) ↓ | Hetero nav. (6v0) ↓ | Phy decep. (4v2) ↓ | Pred-prey (4v4) ↑ | Keep-away (4v4) ↓ |
| **Partial observability** | SPARSE | $0.22 \pm 0.01$ | $0.27 \pm 0.05$ | $0.23 \pm 0.02$ | $2.03 \pm 0.15$ | $2.97 \pm 0.17$ |
| | $CURIO_{self}$ | $0.30 \pm 0.02$ | $0.21 \pm 0.01$ | $0.24 \pm 0.02$ | $2.18 \pm 0.13$ | $2.70 \pm 0.25$ |
| | $CURIO_{team}$ | $0.23 \pm 0.02$ | $0.22 \pm 0.01$ | $0.23 \pm 0.02$ | $2.29 \pm 0.12$ | $3.14 \pm 0.08$ |
| | $ELIGN_{self}$ | $0.29 \pm 0.03$ | $0.19 \pm 0.00$ | $0.24 \pm 0.02$ | $2.39 \pm 0.11$ | $2.97 \pm 0.30$ |
| | $ELIGN_{team}$ | $0.23 \pm 0.04$ | $0.21 \pm 0.00$ | $0.23 \pm 0.01$ | $2.16 \pm 0.12$ | $2.88 \pm 0.19$ |
| | $ELIGN_{adv}$ | — | — | $0.22 \pm 0.01$ | $2.12 \pm 0.16$ | $2.66 \pm 0.23$ |
| **Full observability** | SPARSE | $0.23 \pm 0.06$ | $0.27 \pm 0.05$ | $0.21 \pm 0.02$ | $1.64 \pm 0.02$ | $3.28 \pm 0.02$ |
| | $CURIO_{self}$ | $0.33 \pm 0.09$ | $0.21 \pm 0.01$ | $0.22 \pm 0.01$ | $1.81 \pm 0.12$ | $3.24 \pm 0.08$ |
| | $ELIGN_{self}$ | $0.20 \pm 0.04$ | $0.21 \pm 0.00$ | $0.21 \pm 0.01$ | $1.82 \pm 0.10$ | $2.97 \pm 0.17$ |

Table 7: We report the mean test episode extrinsic rewards and standard errors of *decentralized* methods with different intrinsic rewards in *symmetry-breaking* settings under partial and full observability.

| Task (Agt # vs. Adv #) | | Cooperative | | Competitive | | |
|---|---|---|---|---|---|---|
| | | Coop nav. (3v0) | Hetero nav. (4v0) | Phy decep. (2v1) | Pred-prey (2v2) | Keep-away (2v2) |
| **Partial observability** | SPARSE | $97.45 \pm 10.49$ | $184.18 \pm 7.63$ | $59.39 \pm 21.10$ | $-1.89 \pm 1.69$ | $3.85 \pm 4.25$ |
| | $CURIO_{self}$ | $85.23 \pm 10.88$ | $184.07 \pm 9.99$ | $54.17 \pm 27.40$ | $-2.86 \pm 1.19$ | $19.57 \pm 4.92$ |
| | $CURIO_{team}$ | $81.50 \pm 15.78$ | $141.78 \pm 20.04$ | $41.12 \pm 13.37$ | $-2.80 \pm 1.91$ | $10.21 \pm 6.34$ |
| | $ELIGN_{self}$ | $110.29 \pm 9.67$ | $176.98 \pm 6.38$ | $98.90 \pm 17.71$ | $-4.00 \pm 2.14$ | $9.47 \pm 3.99$ |
| | $ELIGN_{team}$ | $92.41 \pm 10.70$ | $187.42 \pm 11.29$ | $74.06 \pm 21.58$ | $-2.00 \pm 1.39$ | $3.32 \pm 3.04$ |
| | $ELIGN_{adv}$ | — | — | $87.55 \pm 15.35$ | $-1.40 \pm 1.25$ | $13.77 \pm 3.58$ |
| **Full observability** | SPARSE | $150.42 \pm 15.18$ | $250.41 \pm 14.23$ | $69.06 \pm 14.06$ | $-7.62 \pm 3.50$ | $3.50 \pm 4.00$ |
| | $CURIO_{self}$ | $149.48 \pm 9.42$ | $241.69 \pm 19.58$ | $52.69 \pm 17.97$ | $-10.40 \pm 6.33$ | $-1.10 \pm 0.59$ |
| | $ELIGN_{self}$ | $152.08 \pm 6.68$ | $275.69 \pm 7.49$ | $75.79 \pm 24.54$ | $-4.44 \pm 2.05$ | $0.96 \pm 3.14$ |

Table 8: The average test occupancy/collision count per step and standard errors of *decentralized* methods with different intrinsic rewards in *symmetry-breaking* settings under partial and full observability. Higher scores are better for the occupancy metric (↑), and lower scores are better for the collision metric (↓).

| Task (Agt # vs. Adv #) | | Cooperative | | Competitive | | |
| --- | --- | --- | --- | --- | --- | --- |
| | | Coop nav. (3v0) ↑ | Hetero nav. (4v0) ↑ | Phy decep. (2v1) ↑ | Pred-prey (2v2) ↓ | Keep-away (2v2) ↑ |
| **Partial observability** | SPARSE | $0.26 \pm 0.04$ | $0.27 \pm 0.02$ | $0.67 \pm 0.09$ | $0.02 \pm 0.01$ | $0.04 \pm 0.03$ |
| | CURIO$_{self}$ | $0.22 \pm 0.01$ | $0.28 \pm 0.04$ | $0.61 \pm 0.06$ | $0.02 \pm 0.01$ | $0.15 \pm 0.04$ |
| | CURIO$_{team}$ | $0.26 \pm 0.06$ | $0.29 \pm 0.02$ | $0.65 \pm 0.05$ | $0.01 \pm 0.01$ | $0.08 \pm 0.04$ |
| | ELIGN$_{self}$ | $0.29 \pm 0.05$ | $0.32 \pm 0.03$ | $0.62 \pm 0.02$ | $0.02 \pm 0.01$ | $0.08 \pm 0.03$ |
| | ELIGN$_{team}$ | $0.27 \pm 0.04$ | $0.27 \pm 0.02$ | $0.72 \pm 0.10$ | $0.02 \pm 0.01$ | $0.05 \pm 0.04$ |
| | ELIGN$_{adv}$ | — | — | $0.68 \pm 0.07$ | $0.00 \pm 0.00$ | $0.14 \pm 0.04$ |
| **Full observability** | SPARSE | $0.45 \pm 0.12$ | $0.54 \pm 0.01$ | $0.89 \pm 0.11$ | $0.03 \pm 0.01$ | $0.05 \pm 0.02$ |
| | CURIO$_{self}$ | $0.48 \pm 0.08$ | $0.54 \pm 0.01$ | $1.13 \pm 0.14$ | $0.04 \pm 0.02$ | $0.00 \pm 0.00$ |
| | ELIGN$_{self}$ | $0.46 \pm 0.11$ | $0.54 \pm 0.01$ | $0.86 \pm 0.12$ | $0.02 \pm 0.01$ | $0.02 \pm 0.02$ |

Table 9: The average test agent-to-target (agt-target) and agent-to-adversary (agt-adv) distances and standard errors of *decentralized* methods with different intrinsic rewards in *symmetry-breaking* settings under partial and full observability. Lower scores are better for agt-target (↓), and higher scores are better for agt-adv (↑).

| Task (Agt # vs. Adv #) | | Cooperative | | Competitive | | |
| --- | --- | --- | --- | --- | --- | --- |
| | | Coop nav. (3v0) ↓ | Hetero nav. (4v0) ↓ | Phy decep. (2v1) ↓ | Pred-prey (2v2) ↑ | Keep-away (2v2) ↓ |
| **Partial observability** | SPARSE | $0.53 \pm 0.02$ | $0.57 \pm 0.02$ | $0.35 \pm 0.03$ | $1.49 \pm 0.14$ | $1.57 \pm 0.15$ |
| | CURIO$_{self}$ | $0.57 \pm 0.04$ | $0.55 \pm 0.04$ | $0.37 \pm 0.02$ | $1.29 \pm 0.06$ | $1.07 \pm 0.17$ |
| | CURIO$_{team}$ | $0.53 \pm 0.03$ | $0.55 \pm 0.03$ | $0.35 \pm 0.02$ | $1.49 \pm 0.13$ | $1.37 \pm 0.18$ |
| | ELIGN$_{self}$ | $0.68 \pm 0.05$ | $0.52 \pm 0.03$ | $0.34 \pm 0.01$ | $1.39 \pm 0.13$ | $1.26 \pm 0.18$ |
| | ELIGN$_{team}$ | $0.55 \pm 0.05$ | $0.56 \pm 0.02$ | $0.31 \pm 0.02$ | $1.41 \pm 0.10$ | $1.53 \pm 0.17$ |
| | ELIGN$_{adv}$ | — | — | $0.33 \pm 0.04$ | $1.61 \pm 0.08$ | $1.08 \pm 0.15$ |
| **Full observability** | SPARSE | $0.45 \pm 0.12$ | $0.29 \pm 0.00$ | $0.25 \pm 0.01$ | $1.28 \pm 0.15$ | $1.30 \pm 0.15$ |
| | CURIO$_{self}$ | $0.33 \pm 0.06$ | $0.30 \pm 0.01$ | $0.22 \pm 0.01$ | $1.47 \pm 0.17$ | $1.71 \pm 0.05$ |
| | ELIGN$_{self}$ | $0.46 \pm 0.11$ | $0.30 \pm 0.00$ | $0.25 \pm 0.02$ | $1.49 \pm 0.16$ | $1.53 \pm 0.16$ |

Table 10: We report the mean test episode extrinsic rewards and standard errors of *decentralized* methods with different intrinsic rewards in *scaled* and *symmetry-breaking* settings.

| Task (Agt # vs. Adv #) | | Cooperative | | Competitive | | |
| --- | --- | --- | --- | --- | --- | --- |
| | | Coop nav. (5v0) | Hetero nav. (6v0) | Phy decep. (4v2) | Pred-prey (4v4) | Keep-away (4v4) |
| **Partial observability** | SPARSE | $328.24 \pm 24.17$ | $405.08 \pm 21.53$ | $172.87 \pm 32.43$ | $-35.40 \pm 8.63$ | $1.37 \pm 3.48$ |
| | CURIO$_{self}$ | $295.48 \pm 20.54$ | $436.17 \pm 26.30$ | $202.39 \pm 26.06$ | $-11.19 \pm 3.65$ | $9.24 \pm 8.49$ |
| | CURIO$_{team}$ | $316.33 \pm 14.44$ | $422.71 \pm 13.24$ | $229.50 \pm 28.29$ | $-11.56 \pm 6.37$ | $-1.29 \pm 1.58$ |
| | ELIGN$_{self}$ | $357.40 \pm 19.52$ | $412.39 \pm 12.63$ | $129.07 \pm 51.08$ | $-7.34 \pm 5.12$ | $11.97 \pm 13.30$ |
| | ELIGN$_{team}$ | $354.14 \pm 19.53$ | $417.94 \pm 22.29$ | $184.21 \pm 23.16$ | $-19.37 \pm 6.44$ | $4.05 \pm 5.78$ |
| | ELIGN$_{adv}$ | — | — | $148.69 \pm 31.79$ | $-23.42 \pm 8.32$ | $18.71 \pm 14.78$ |
| **Full observability** | SPARSE | $466.17 \pm 28.16$ | $471.19 \pm 16.23$ | $233.61 \pm 25.44$ | $-39.24 \pm 6.63$ | $-5.10 \pm 0.26$ |
| | CURIO$_{self}$ | $509.91 \pm 14.10$ | $606.07 \pm 7.55$ | $256.13 \pm 41.13$ | $-38.66 \pm 13.38$ | $-6.58 \pm 1.29$ |
| | ELIGN$_{self}$ | $520.25 \pm 9.68$ | $510.18 \pm 25.71$ | $222.31 \pm 15.39$ | $-30.56 \pm 9.87$ | $-4.27 \pm 2.53$ |

Table 11: The average test occupancy/collision count per step and standard errors of *decentralized* methods with different intrinsic rewards in *scaled* and *symmetry-breaking* settings. under partial and full observability. Higher scores are better for the occupancy metric (↑), and lower scores are better for the collision metric (↓).

| Task (Agt # vs. Adv #) | | Cooperative | | Competitive | | |
| --- | --- | --- | --- | --- | --- | --- |
| | | Coop nav. (5v0) ↑ | Hetero nav. (6v0) ↑ | Phy decep. (4v2) ↑ | Pred-prey (4v4) ↓ | Keep-away (4v4) ↑ |
| **Partial observability** | SPARSE | $0.37 \pm 0.05$ | $0.38 \pm 0.03$ | $0.75 \pm 0.04$ | $0.13 \pm 0.03$ | $0.04 \pm 0.02$ |
| | CURIO$_{self}$ | $0.29 \pm 0.03$ | $0.43 \pm 0.02$ | $0.66 \pm 0.06$ | $0.05 \pm 0.02$ | $0.12 \pm 0.08$ |
| | CURIO$_{team}$ | $0.32 \pm 0.02$ | $0.34 \pm 0.01$ | $0.78 \pm 0.07$ | $0.05 \pm 0.02$ | $0.02 \pm 0.01$ |
| | ELIGN$_{self}$ | $0.29 \pm 0.03$ | $0.39 \pm 0.02$ | $0.96 \pm 0.20$ | $0.04 \pm 0.03$ | $0.11 \pm 0.10$ |
| | ELIGN$_{team}$ | $0.37 \pm 0.03$ | $0.40 \pm 0.03$ | $0.72 \pm 0.03$ | $0.07 \pm 0.03$ | $0.06 \pm 0.03$ |
| | ELIGN$_{adv}$ | — | — | $0.63 \pm 0.09$ | $0.07 \pm 0.03$ | $0.15 \pm 0.10$ |
| **Full observability** | SPARSE | $0.52 \pm 0.11$ | $0.43 \pm 0.09$ | $0.86 \pm 0.07$ | $0.19 \pm 0.02$ | $0.00 \pm 0.00$ |
| | CURIO$_{self}$ | $0.39 \pm 0.14$ | $0.54 \pm 0.00$ | $0.81 \pm 0.06$ | $0.14 \pm 0.04$ | $0.00 \pm 0.00$ |
| | ELIGN$_{self}$ | $0.55 \pm 0.11$ | $0.55 \pm 0.00$ | $0.94 \pm 0.08$ | $0.12 \pm 0.03$ | $0.02 \pm 0.01$ |

Table 12: The average test agent-to-target (agt-target) and agent-to-adversary (agt-adv) distances and standard errors of *decentralized* methods with different intrinsic rewards in *scaled* and *symmetry-breaking* settings under partial and full observability. Lower scores are better for agt-target (↓), and higher scores are better for agt-adv (↑).

| | | Cooperative | | Competitive | | |
| --- | --- | --- | --- | --- | --- | --- |
| Task (Agt # vs. Adv #) | | Coop nav. (5v0) ↓ | Hetero nav. (6v0) ↓ | Phy decep. (4v2) ↓ | Pred-prey (4v4) ↑ | Keep-away (4v4) ↓ |
| **Partial observability** | SPARSE | $0.36 \pm 0.01$ | $0.42 \pm 0.02$ | $0.35 \pm 0.01$ | $2.04 \pm 0.18$ | $3.10 \pm 0.29$ |
| | $CURIO_{self}$ | $0.50 \pm 0.04$ | $0.37 \pm 0.02$ | $0.38 \pm 0.03$ | $2.35 \pm 0.12$ | $2.70 \pm 0.35$ |
| | $CURIO_{team}$ | $0.43 \pm 0.03$ | $0.45 \pm 0.01$ | $0.35 \pm 0.02$ | $2.39 \pm 0.15$ | $3.37 \pm 0.19$ |
| | $ELIGN_{self}$ | $0.52 \pm 0.05$ | $0.41 \pm 0.02$ | $0.37 \pm 0.04$ | $2.52 \pm 0.15$ | $3.22 \pm 0.42$ |
| | $ELIGN_{team}$ | $0.42 \pm 0.03$ | $0.41 \pm 0.02$ | $0.37 \pm 0.01$ | $2.25 \pm 0.15$ | $3.00 \pm 0.33$ |
| | $ELIGN_{adv}$ | — | — | $0.41 \pm 0.05$ | $2.27 \pm 0.16$ | $2.62 \pm 0.30$ |
| **Full observability** | SPARSE | $0.29 \pm 0.07$ | $0.37 \pm 0.06$ | $0.26 \pm 0.01$ | $1.81 \pm 0.04$ | $3.69 \pm 0.05$ |
| | $CURIO_{self}$ | $0.42 \pm 0.11$ | $0.29 \pm 0.00$ | $0.26 \pm 0.02$ | $2.02 \pm 0.16$ | $3.63 \pm 0.12$ |
| | $ELIGN_{self}$ | $0.26 \pm 0.05$ | $0.29 \pm 0.00$ | $0.27 \pm 0.01$ | $2.10 \pm 0.12$ | $3.24 \pm 0.26$ |

Table 13: We report the mean test episode extrinsic rewards and standard errors of *centralized* methods with different intrinsic rewards under full observability.

| | | Cooperative | | Competitive | | |
| --- | --- | --- | --- | --- | --- | --- |
| Task (Agt # vs. Adv #) | | Coop nav. (3v0) | Hetero nav. (4v0) | Phy decep. (2v1) | Pred-prey (2v2) | Keep-away (2v2) |
| **Full observability** | SPARSE | $106.02 \pm 20.95$ | $123.17 \pm 18.77$ | $130.90 \pm 6.59$ | $-1.90 \pm 1.61$ | $12.49 \pm 9.83$ |
| | $CURIO_{self}$ | $86.52 \pm 16.02$ | $108.84 \pm 6.89$ | $107.84 \pm 13.67$ | $-1.69 \pm 0.60$ | $23.70 \pm 12.95$ |
| | $ELIGN_{self}$ | $120.47 \pm 12.26$ | $134.30 \pm 5.84$ | $105.74 \pm 9.72$ | $-2.37 \pm 1.39$ | $22.92 \pm 7.00$ |

Table 14: The average test occupancy/collision count per step and standard errors of *centralized* methods with different intrinsic rewards under partial and full observability. Higher scores are better for the occupancy metric (↑), and lower scores are better for the collision metric (↓).

| | | Cooperative | | Competitive | | |
| --- | --- | --- | --- | --- | --- | --- |
| Task (Agt # vs. Adv #) | | Coop nav. (3v0) ↑ | Hetero nav. (4v0) ↑ | Phy decep. (2v1) ↑ | Pred-prey (2v2) ↓ | Keep-away (2v2) ↑ |
| **Partial observability** | SPARSE | $0.29 \pm 0.10$ | $0.50 \pm 0.03$ | $0.94 \pm 0.06$ | $0.00 \pm 0.00$ | $0.36 \pm 0.12$ |
| | $CURIO_{self}$ | $0.28 \pm 0.09$ | $0.47 \pm 0.03$ | $0.94 \pm 0.03$ | $0.01 \pm 0.00$ | $0.17 \pm 0.10$ |
| | $CURIO_{team}$ | $0.33 \pm 0.10$ | $0.47 \pm 0.04$ | $0.92 \pm 0.01$ | $0.01 \pm 0.00$ | $0.08 \pm 0.03$ |
| | $ELIGN_{self}$ | $0.21 \pm 0.10$ | $0.50 \pm 0.01$ | $0.92 \pm 0.02$ | $0.00 \pm 0.00$ | $0.25 \pm 0.08$ |
| | $ELIGN_{team}$ | $0.23 \pm 0.09$ | $0.55 \pm 0.02$ | $0.90 \pm 0.07$ | $0.01 \pm 0.00$ | $0.24 \pm 0.11$ |
| | $ELIGN_{adv}$ | — | — | $0.94 \pm 0.04$ | $0.00 \pm 0.00$ | $0.10 \pm 0.05$ |
| **Full observability** | SPARSE | $0.34 \pm 0.10$ | $0.33 \pm 0.07$ | $0.88 \pm 0.04$ | $0.01 \pm 0.00$ | $0.26 \pm 0.11$ |
| | $CURIO_{self}$ | $0.30 \pm 0.07$ | $0.32 \pm 0.05$ | $0.82 \pm 0.02$ | $0.01 \pm 0.01$ | $0.33 \pm 0.16$ |
| | $ELIGN_{self}$ | $0.30 \pm 0.11$ | $0.40 \pm 0.04$ | $0.88 \pm 0.05$ | $0.01 \pm 0.01$ | $0.30 \pm 0.07$ |

Table 15: The average test agent-to-target (agt-target) and agent-to-adversary (agt-adv) distances and standard errors of *centralized* methods with different intrinsic rewards under partial and full observability. Lower scores are better for agt-target (↓), and higher scores are better for agt-adv (↑).

| | | Cooperative | | Competitive | | |
| --- | --- | --- | --- | --- | --- | --- |
| Task (Agt # vs. Adv #) | | Coop nav. (3v0) ↓ | Hetero nav. (4v0) ↓ | Phy decep. (2v1) ↓ | Pred-prey (2v2) ↑ | Keep-away (2v2) ↓ |
| **Partial observability** | SPARSE | $0.42 \pm 0.05$ | $0.29 \pm 0.02$ | $0.27 \pm 0.01$ | $1.54 \pm 0.02$ | $1.38 \pm 0.13$ |
| | $CURIO_{self}$ | $0.42 \pm 0.05$ | $0.29 \pm 0.01$ | $0.27 \pm 0.01$ | $1.46 \pm 0.05$ | $1.40 \pm 0.13$ |
| | $CURIO_{team}$ | $0.41 \pm 0.06$ | $0.29 \pm 0.02$ | $0.28 \pm 0.01$ | $1.49 \pm 0.04$ | $1.43 \pm 0.14$ |
| | $ELIGN_{self}$ | $0.50 \pm 0.07$ | $0.29 \pm 0.01$ | $0.27 \pm 0.01$ | $1.60 \pm 0.04$ | $1.26 \pm 0.12$ |
| | $ELIGN_{team}$ | $0.45 \pm 0.05$ | $0.27 \pm 0.01$ | $0.28 \pm 0.01$ | $1.52 \pm 0.04$ | $1.35 \pm 0.14$ |
| | $ELIGN_{adv}$ | — | — | $0.28 \pm 0.01$ | $1.55 \pm 0.03$ | $1.45 \pm 0.10$ |
| **Full observability** | SPARSE | $0.38 \pm 0.07$ | $0.34 \pm 0.04$ | $0.25 \pm 0.00$ | $1.59 \pm 0.06$ | $1.43 \pm 0.09$ |
| | $CURIO_{self}$ | $0.36 \pm 0.05$ | $0.32 \pm 0.02$ | $0.25 \pm 0.01$ | $1.53 \pm 0.09$ | $1.08 \pm 0.15$ |
| | $ELIGN_{self}$ | $0.43 \pm 0.08$ | $0.30 \pm 0.02$ | $0.25 \pm 0.00$ | $1.51 \pm 0.08$ | $1.18 \pm 0.15$ |

Table 16: We report the mean test episode extrinsic rewards and standard errors of *centralized* methods with different intrinsic rewards in *scaled* environments.

| | | Cooperative | | Competitive | | |
| --- | --- | --- | --- | --- | --- | --- |
| Task (Agt # vs. Adv #) | | Coop nav. (5v0) | Hetero nav. (6v0) | Phy decep. (4v2) | Pred-prey (4v4) | Keep-away (4v4) |
| **Partial observability** | SPARSE | $100.63 \pm 19.36$ | $346.16 \pm 18.95$ | $-38.99 \pm 16.18$ | $-17.33 \pm 4.29$ | $-2.50 \pm 2.64$ |
| | $ELIGN_{self}$ | $112.15 \pm 19.69$ | $375.21 \pm 26.10$ | $13.71 \pm 29.53$ | $-20.12 \pm 1.42$ | $-4.68 \pm 1.21$ |
| | $ELIGN_{team}$ | $97.93 \pm 25.23$ | $372.41 \pm 44.28$ | $60.07 \pm 13.26$ | $-27.87 \pm 0.99$ | $1.72 \pm 3.79$ |
| | $ELIGN_{adv}$ | — | — | $21.67 \pm 48.17$ | $-17.68 \pm 5.59$ | $-4.92 \pm 1.81$ |
| **Full observability** | SPARSE | $50.60 \pm 13.10$ | $153.76 \pm 19.81$ | $97.32 \pm 17.95$ | $-38.25 \pm 5.06$ | $-3.39 \pm 2.77$ |
| | $ELIGN_{self}$ | $186.55 \pm 53.15$ | $127.97 \pm 13.02$ | $103.46 \pm 28.91$ | $-23.29 \pm 5.00$ | $-4.90 \pm 0.67$ |

Table 17: The average test occupancy/collision count per step and standard errors of *centralized* methods with different intrinsic rewards in *scaled* environments under partial and full observability. Higher scores are better for the occupancy metric (↑), and lower scores are better for the collision metric (↓).

| Task (Agt # vs. Adv #) | | Cooperative | | Competitive | | |
| | | Coop nav. (5v0) ↑ | Hetero nav. (6v0) ↑ | Phy decep. (4v2) ↑ | Pred-prey (4v4) ↓ | Keep-away (4v4) ↑ |
|---|---|---|---|---|---|---|
| **Partial observability** | SPARSE | $0.11 \pm 0.02$ | $0.29 \pm 0.06$ | $0.56 \pm 0.05$ | $0.06 \pm 0.02$ | $0.07 \pm 0.02$ |
| | $\text{ELIGN}_{\text{self}}$ | $0.23 \pm 0.09$ | $0.33 \pm 0.04$ | $0.56 \pm 0.06$ | $0.08 \pm 0.00$ | $0.05 \pm 0.00$ |
| | $\text{ELIGN}_{\text{team}}$ | $0.27 \pm 0.10$ | $0.33 \pm 0.05$ | $0.50 \pm 0.08$ | $0.09 \pm 0.00$ | $0.09 \pm 0.03$ |
| | $\text{ELIGN}_{\text{adv}}$ | — | — | $0.60 \pm 0.09$ | $0.05 \pm 0.02$ | $0.05 \pm 0.01$ |
| **Full observability** | SPARSE | $0.10 \pm 0.04$ | $0.16 \pm 0.04$ | $0.50 \pm 0.03$ | $0.12 \pm 0.01$ | $0.06 \pm 0.01$ |
| | $\text{ELIGN}_{\text{self}}$ | $0.16 \pm 0.09$ | $0.11 \pm 0.00$ | $0.55 \pm 0.02$ | $0.10 \pm 0.02$ | $0.04 \pm 0.01$ |

Table 18: The average test agent-to-target (agt-target) and agent-to-adversary (agt-adv) distances and standard errors of *centralized* methods with different intrinsic rewards in *scaled* environments under partial and full observability. Lower scores are better for agt-target (↓), and higher scores are better for agt-adv (↑).

| Task (Agt # vs. Adv #) | | Cooperative | | Competitive | | |
| | | Coop nav. (5v0) ↓ | Hetero nav. (6v0) ↓ | Phy decep. (4v2) ↓ | Pred-prey (4v4) ↑ | Keep-away (4v4) ↓ |
|---|---|---|---|---|---|---|
| **Partial observability** | SPARSE | $0.33 \pm 0.01$ | $0.29 \pm 0.02$ | $0.34 \pm 0.02$ | $2.27 \pm 0.08$ | $3.12 \pm 0.18$ |
| | $\text{ELIGN}_{\text{self}}$ | $0.32 \pm 0.04$ | $0.28 \pm 0.01$ | $0.36 \pm 0.02$ | $2.32 \pm 0.09$ | $3.25 \pm 0.05$ |
| | $\text{ELIGN}_{\text{team}}$ | $0.30 \pm 0.04$ | $0.28 \pm 0.01$ | $0.37 \pm 0.03$ | $2.29 \pm 0.08$ | $3.01 \pm 0.20$ |
| | $\text{ELIGN}_{\text{adv}}$ | — | — | $0.33 \pm 0.04$ | $2.44 \pm 0.08$ | $3.23 \pm 0.13$ |
| **Full observability** | SPARSE | $0.37 \pm 0.03$ | $0.37 \pm 0.02$ | $0.29 \pm 0.02$ | $1.98 \pm 0.07$ | $3.13 \pm 0.17$ |
| | $\text{ELIGN}_{\text{self}}$ | $0.36 \pm 0.04$ | $0.39 \pm 0.00$ | $0.27 \pm 0.01$ | $1.98 \pm 0.08$ | $3.25 \pm 0.12$ |

Table 19: We sample agents from different *decentralized* training runs and evaluate their zero-shot performance in *scaled* environments under partial observability. We report the mean test episode extrinsic rewards and standard errors of decentralized methods with different intrinsic rewards.

| Task (Agt # vs. Adv #) | | Cooperative | | Competitive | | |
| | | Coop nav. (5v0) | Hetero nav. (6v0) | Phy decep. (4v2) | Pred-prey (4v4) | Keep-away (4v4) |
|---|---|---|---|---|---|---|
| **Partial observability** | SPARSE | $434.68 \pm 6.42$ | $561.16 \pm 31.63$ | $128.64 \pm 17.31$ | $-32.12 \pm 3.63$ | $-2.80 \pm 2.91$ |
| | $\text{ELIGN}_{\text{self}}$ | $471.07 \pm 5.00$ | $676.01 \pm 16.53$ | $248.16 \pm 6.62$ | $-16.77 \pm 2.25$ | $-5.03 \pm 1.06$ |
| | $\text{ELIGN}_{\text{team}}$ | $511.97 \pm 6.95$ | $699.56 \pm 11.64$ | $190.06 \pm 29.10$ | $-19.40 \pm 2.86$ | $-3.10 \pm 3.09$ |
| | $\text{ELIGN}_{\text{adv}}$ | — | — | $228.53 \pm 25.03$ | $-31.03 \pm 3.13$ | $27.24 \pm 4.48$ |

Table 20: We sample agents from different *decentralized* training runs and evaluate their zero-shot performance in *scaled* environments under partial observability. We report the average test agent-to-target (agt-target) and agent-to-adversary (agt-adv) distances and standard errors of *decentralized* methods with different intrinsic rewards. Lower scores are better for agt-target (↓), and higher scores are better for agt-adv (↑).

| Task (Agt # vs. Adv #) | | Cooperative | | Competitive | | |
| | | Coop nav. (5v0) ↓ | Hetero nav. (6v0) ↓ | Phy decep. (4v2) ↓ | Pred-prey (4v4) ↑ | Keep-away (4v4) ↓ |
|---|---|---|---|---|---|---|
| **Partial observability** | SPARSE | $0.22 \pm 0.00$ | $0.23 \pm 0.00$ | $0.23 \pm 0.00$ | $1.93 \pm 0.00$ | $3.16 \pm 0.01$ |
| | $\text{ELIGN}_{\text{self}}$ | $0.19 \pm 0.00$ | $0.20 \pm 0.00$ | $0.17 \pm 0.00$ | $2.33 \pm 0.00$ | $3.31 \pm 0.01$ |
| | $\text{ELIGN}_{\text{team}}$ | $0.43 \pm 0.01$ | $0.19 \pm 0.00$ | $0.24 \pm 0.00$ | $2.04 \pm 0.01$ | $3.15 \pm 0.01$ |
| | $\text{ELIGN}_{\text{adv}}$ | — | — | $0.21 \pm 0.00$ | $2.11 \pm 0.01$ | $2.31 \pm 0.01$ |

Table 21: Model and training hyperparameters

| Parameter | Multi-agent particle | Google Research football |
|---|---|---|
| SAC actor model architecture | FC layers [128,128] | FC layers [256,256] |
| SAC critic model architecture | FC layers [128,128] | FC layers [256,256] |
| World model architecture | FC layers [128,128] | FC layers [128,128] |
| Replay buffer size | 1,000,000 | 1,000,000 |
| Batch size | 1,024 | 256 |
| Actor learning rate | 0.001 | 0.0003 |
| Critic learning rate | 0.001 | 0.0003 |
| Discount factor gamma | 0.95 | 0.99 |
| SAC soft update coefficient | 0.01 | 0.005 |
| SAC policy entropy regularization coefficient | 0.1 | 1.0 (initial) |