# OpenReview forum: "ELIGN: Expectation Alignment as a Multi-Agent Intrinsic Reward"
_NeurIPS.cc/2022/Conference — NeurIPS 2022 Accept_

### Official Review · Reviewer_TEM3 · 2022-07-03

**Rating:** 4
**Confidence:** 3
**Soundness:** 2 fair
**Presentation:** 4 excellent
**Contribution:** 2 fair

**Summary:**

This paper studies a new self-supervised MARL intrinsic reward, called Alignment, to facilitate learning in a decentralized training paradigm under partial observability. Inspired by the self-organization principle in Zoology, the intrinsic reward of Alignment aims to guide agents to match their teammates' expectations in order to emerge cooperative strategies. The experiments are carried out on the multi-agent particle and Google Research football, and the results show that Alignment outperforms sparse and curiosity-based intrinsic rewards.


**Questions:**

The following questions correspond to three points listed in the WEAKNESSES.

1) How do the agents share their expectations with teammates during decentralized training?

2) When the teammates have entirely different expectations for agent $i$, how does agent $i$ match their expectations?

3) Why use tables instead of learning curves to visualize learning performance?

**Limitations:**

The main concerns are listed in the WEAKNESSES. The authors should justify the soundness of the proposed method in Section 4.2.

**Strengths And Weaknesses:**

The reviewer will list the main strengths and weaknesses as follows:

STRENGTHS

1) This paper introduces an interesting intrinsic reward, called Alignment, in decentralized training under partial observability. This idea is inspired by Zoology and brings a new heuristic to emerge coordination during decentralized training.

2) This paper conducts several experiments on popular MARL benchmarks to show that the Alignment intrinsic reward is a more useful method than curiosity-driven exploration.

WEAKNESSES

1) The major concern is that the method of Alignment (see Section 4) theoretically conflicts with the decentralized training paradigm. In decentralized training, agents cannot share their expectations. Thus, each agent theoretically cannot match their teammates' expectations because they do not know them. For example, the dynamics model of agent $i$ in line 150 is trained on the $o_i$ and $a_i$ but the alignment intrinsic reward $r_{in}$ defined in line 155 apply the dynamics model of agent $j$ on the $o_i$ and $a_i$. Note that the dynamics model of agent $j$ is trained on the set of $o_j, a_j$ rather than $o_i,a_i$. $r_{in}$ defined in line 155 may induce huge penalty due to this distributional shift from $o_j,a_j$ to $o_i,a_i$.

2) When the agent aims to match their teammates' expectations, what if their teammates do not reach a consensus? For example, in the definition of $r_{in}$ on line 155, if agent $i$'s neighbors have completely different expectations, which expectation does agent $i$ match?

3) Sample efficiency is an important measure for MARL evaluation. Could you provide the learning curve during training to better visualize the sample efficiency in comparison? Tables may not be a natural way to demonstrate the advantage of exploration methods.

---

> ### Author Response · Authors · 2022-08-02
> **We have revised our paper to improve clarity on our method and included the plots suggested by the reviewer.**
>
> We thank the reviewer for their feedback and questions.
>
> > The method of Alignment (see Section 4) theoretically conflicts with the decentralized training paradigm…How do the agents share their expectations with teammates during decentralized training?
>
> Sorry for the misunderstanding. In line 159, we point out that in decentralized training, agent $i$ uses its own dynamics model $f_{\theta_i}$ to approximate its teammate agent $j$’s dynamics. Therefore, our alignment reward does not conflict with decentralized training. We have updated the text so that this is more salient.
>
> One might ask whether this proxy is good enough to imitate other agents. We answer this in the position with the following experiment: We compare ALIGNteam with the ground truth dynamics model and the proxy model in Cooperative navigation (3v0), and we find that the former achieves a test episode reward of 147.83 ± 14.27 compared to 141.04 ± 8.04, suggesting that our assumption of using an agent’s own dynamics model is a good proxy.
>
> > When the teammates have entirely different expectations for agent $i$, how does agent $i$  match their expectations?
>
> Thank you for this question. We ran an experiment on cooperative and heterogeneous navigation where we tracked the difference in expectations across agents. Our experiments indicate that the difference decreases over training. The results suggest that teammates do have different expectations, however over training they predict each other’s dynamics better. During inference, there can still be scenarios in which expectations from two neighbors might diverge. But since the expectations are only used to tune the intrinsic reward during training, and not used during inference, a divergence in expectations should not impact test time performance.
>
> > Why use tables instead of learning curves to visualize learning performance?
>
> Thanks for the suggestion. We have now included learning curves to visualize the sample efficiency of different methods. Compared to the best baseline’s highest performance at the 100th epoch, we find that it only takes the best alignment variant 31 (Coop nav), 74 (Hetero nav), 30 (Physical dec), 97 (Predator-prey) and 92 (Keep-away) epochs respectively in the five multi-agent particle tasks. This means that on average the best alignment variant requires $35\pm 32$ fewer training steps to reach the same performance as curiosity or spare methods.

---

### Official Review · Reviewer_DoA9 · 2022-07-08

**Rating:** 7
**Confidence:** 4
**Soundness:** 4 excellent
**Presentation:** 3 good
**Contribution:** 3 good

**Summary:**

The authors propose an auxiliary training reward called ALIGNment. Agents are rewarded when either they (ALIGNself) or their teammates (ALIGNteam) can predict their actions, or when their adversaries (ALIGNadv) cannot predict their actions. The authors use experimental evidence to support the intuition that alignment helps agents coordinate which subtasks to complete and helps agents coordinate zero-shot with new partners. Compared with 3 prior approaches in 6 Google Research football environments and 5 multi-agent particle environments, at least one ALIGN variant achieves state-of-the-art results in every task except for the heterogeneous navigation task.

**Questions:**

Did you train the Sparse, Curio-self, and Curio-team methods yourself? Or did you take numbers from prior work? It seems like you trained them yourself, but it would be helpful to make this more clear.  (I think it’s stronger if you can report numbers directly from prior work where applicable, because the authors of prior work have an incentive to tune their algorithms as well as possible.)

If you trained prior methods yourself: Did you spend equal effort tuning the prior approaches as you did tuning your own method, and can you quantify this in the paper (eg by having tested the same number of hyperparameter settings)?

**Limitations:**

The authors note that their method, ALIGN, performs worse than prior methods in Heterogenous navigation. They hypothesize that this is because the differing size and speed of agents in Heterogenous navigation makes it difficult for agents to predict each other. I appreciate that the authors analyze this limitation.

**Strengths And Weaknesses:**

## Strengths
Overall, I found the method easy to understand. Algorithm 1 makes the method clear.

The experimental evaluation is thorough, comparing against 3 prior approaches and 11 total tasks. At least one ALIGN variant performs best across all but one task.

The authors include additional experiments to help study why ALIGNment improves performance.

As this is an intuitive approach that is explained clearly and supported by thorough experimental evaluation, I recommend acceptance.

## Weaknesses
The word “alignment” already has a meaning in artificial intelligence. Specifically, it is used to mean how well an artificial intelligence is aligned to human preferences. See, for example, “The Alignment Problem” by Brian Christian. I would recommend the authors use a different word for their method, such as “Predictability.” This renaming would respect the existing use of the word “alignment” as well as be more specific about what the auxiliary loss is rewarding.

I didn’t find Figure 1 very helpful for understanding the method. The idea behind alignment – a predictive loss – is very simple, yet Figure 1 is quite complex. I’d recommend trying to redesign Figure 1 to make it as simple as possible.

The overall results are presented in two dense tables, Table 1 and Table 2. It would be helpful if the authors could present aggregated results in a single figure. If possible, I would recommend normalizing results using the performance of an expert policy and a random policy as is done in the D4RL benchmark [1], i.e., computing (performance - random) / (expert - random). This gives a common scale over which the authors can aggregate.

Only the maximum performance is bolded in the results tables. I recommend bolding all results within 5% or 10% of maximum performance, to account for noise, as is done in other work such as [2].

[1] D4RL: Datasets for Deep Data-Driven Reinforcement Learning

[2] Offline Reinforcement Learning with Implicit Q-Learning

---

> ### Author Response · Authors · 2022-08-02
> **We appreciate your positive comments and thoughtful suggestions.**
>
> > The word “alignment” already has a meaning in artificial intelligence. Specifically, it is used to mean how well an artificial intelligence is aligned to human preferences. See, for example, “The Alignment Problem” by Brian Christian. I would recommend the authors use a different word for their method, such as “Predictability.”
>
> We agree that the name is not perfect. Choosing a unique yet informative name for our method has been more challenging than expected. We are open to your suggestions.
>
> We used alignment because the Zoology literature refers to the phenomenon as “aligning” behavior. We considered synonyms like “comply” or “conform” but those come with negative connotations. We could call our method “align” instead of “alignment” to differentiate it from Brian Christian’s book. Open to your suggestions.
>
> > I didn’t find Figure 1 very helpful for understanding the method. The idea behind alignment – a predictive loss – is very simple, yet Figure 1 is quite complex. I’d recommend trying to redesign Figure 1 to make it as simple as possible.
>
> We have made a minor modification to Figure 1 to make the predictive process used to calculate the alignment reward more intuitive. We also tried to not change it too much since reviewers DoA9 and oB8u mention the figure positively. We are happy to incorporate more changes you suggest.
>
> > The overall results are presented in two dense tables, Table 1 and Table 2. It would be helpful if the authors could present aggregated results in a single figure.
>
> We appreciate the suggestion. We will add such a figure for the camera ready. Although dense, we followed the common precedence set by related works, who present raw numbers in tables, including our baselines (Stadie et al. 2015, Iqbal & Sha 2020).
>
> We also have figures for comparing metrics that are on a similar scale (Figures 2, 5, 6).
>
> We have additionally included learning curves to the appendix.
>
> > Did you train the Sparse, Curio-self, and Curio-team methods yourself? Or did you take numbers from prior work? It seems like you trained them yourself, but it would be helpful to make this more clear. (I think it’s stronger if you can report numbers directly from prior work where applicable, because the authors of prior work have an incentive to tune their algorithms as well as possible.)
>
> Yes, we trained all the baseline methods by ourselves. Unfortunately, all the baseline intrinsic reward methods were trained using different algorithms: Lowe et al. 2007 used MADDPG, Stadie et al. 2015 trained using DQN, Iqbal et al. 2019 extended soft-actor critic to a multi-agent formulation. We chose to train baselines by ourselves in order to standardize compute and training algorithm.
>
> > If you trained prior methods yourself: Did you spend equal effort tuning the prior approaches as you did tuning your own method, and can you quantify this in the paper (eg by having tested the same number of hyperparameter settings)?
>
> For both the baseline methods and for our alignment variants, we ran a hyperparameter sweep with the same value ranges. Section 5.3 describes how we chose the optimal model / hyperparameters. We did not spend additional efforts to improve our method.

---

> > ### Comment · Reviewer_DoA9 · 2022-08-05
> > **Alternative Naming Suggestions**
> >
> > Thanks for your answers to my questions! It makes sense that you needed to train the baselines yourself so that the intrinsic reward methods had a standardized training algorithm. I'm also glad to see that you ran a hyperparameter sweep with the same value ranges for each method.
> >
> > > We agree that the name is not perfect. Choosing a unique yet informative name for our method has been more challenging than expected. We are open to your suggestions.
> >
> > I appreciate your openness to trying to find a name that doesn't conflict with the existing use of the term "alignment." If you don't like "predictability" (which I think is the most accurate description of what's happening), then I would suggest trying to make a name around the word "synchronization". Perhaps a phrase such as "sync" or "in sync" could be helpful. I think the word "alignment" implies that there's an existing direction that the agents are matching. The word "sync(hronization)" seems more appropriate to me to describe agents coordinating by devising their own protocols.

---

> > > ### Author Response · Authors · 2022-08-09
> > > **Changing "alignment" to "predictability"**
> > >
> > > We ended up considering a number of other options: expectation matching, synchronization, invariability, stability, standardization, conventionality, regularity, conformity, and self-fulfilling prophecy. In the end, "Predictability" does seem like the most accurate. We will replace "alignment" with "predictability" in the paper.

---

### Official Review · Reviewer_oB8u · 2022-07-08

**Rating:** 7
**Confidence:** 3
**Soundness:** 3 good
**Presentation:** 3 good
**Contribution:** 3 good

**Summary:**

- This paper proposes to tackle the problem of multi-agent learning under decentralised training and partial observability through using a novel self-supervised intrinsic reward that encourages agents to behave predictably to their neighbouring teammates and unpredictably to their adversaries.
- A dynamics model (trained on the agent’s own observations as a proxy given training is decentralised), is used to predict future states from other agents’ viewpoints under partial observability. The negative error in this prediction is then used as an intrinsic reward to supplement training in environments with sparse extrinsic rewards.
- The effectiveness of this approach is compared to curiosity driven intrinsic rewards on 6 cooperative and competitive multi-agent tasks. Further experiments are conducted in regimes of full observability, centralised training, and zero shot coordination.

**Questions:**

1. Did you compare this intrinsic reward based approach to other (non intrinsic reward based) approaches to the sparse learning problem (e.g. Ndousse et al 2021)?
    - Some discussion around why the framing as an auxiliary reward instead of an auxiliary loss would be helpful.
2. Under what conditions do we expect alignment/misalignment to work well? A discussion around why we see that ALIGN self, team and adv have varying quality on different tasks would be helpful.
    - For example, why does modelling from the other agent’s point of view, instead of directly predicting your own future state (which incorporates the other agents action) more helpful? When and why is ALIGN_team better than ALIGN_self? It seems that the team set up is just a partially observed view of the self set up, because the agent’s own dynamics model is used.
    - In the adversarial case, it seems that the agent will learn to be unpredictable to itself (as it uses its own dynamics model as a proxy) so it is unclear to me why this would help. Conversely, ALIGN_team and ALIGN_self seem to work better than ALIGN_adv on some competitive tasks indicating that it is sometimes better to be predictable to an adversary which seems counterintuitive.
    - In the first paragraph of Section 4 there is discussion about gaining information outside of the agent’s receptive field through encountering surprising actions from other agents. How does the alignment reward help convey this information? Does this help at execution time?
3. In the centralised regime experiments, did you use the actual dynamics model learnt by each agent to calculate the intrinsic reward or did you still use the individual agents dynamics model as a proxy as in the decentralised case?
4. Do you have any suggestions for how to improve performance using alignment for heterogenous agents?
5. Did you evaluate zero shot coordination with CURIO models as well?

**Limitations:**

- The authors state limitations of their approach, namely:
    - the small action/state spaces of the environments considered.
    - difficulty generalising when there are heterogenous agents due to agents learning only one dynamics model (of themselves). They argue that this enables more scalability, but demonstrate that this leads to decreased performance.
    - The method is reliant on the ability to learn a good dynamics model which may be more difficult in noisy, real-world environments.
- I have addressed the other limitations in the strengths and weaknesses section.

**Strengths And Weaknesses:**

##

- *Originality:*
    - The authors frame their work well in the context of the literature on intrinsic motivation in reinforcement learning, and motivate their approach from the natural world. I appreciate that they explicitly state the differences of their approach to closely related work, e.g. Ndousse et al 2021.
    - Some references need to be corrected:
        - Line 45: “*Only a few attempts explore other forms of multi-agent intrinsic rewards”.*   A more specific, referenced statement would be better here.
        - Line 33: To my knowledge, there is no reference to decentralised training and partial observability in Lowe 2017. MADDPG is a centralised training, decentralised execution algorithm. The authors should reference more appropriate work for that statement on task-specific reward shaping to motivate their work.
- *Quality*:
    - Two standard coordination and competition benchmark environments are used to evaluate their framework. The same architectures and optimisation algorithms are chosen to provide a fair comparison against other intrinsic rewards. I particularly liked the symmetry breaking experiments to demonstrate learned coordination.
    - Table 1 shows promising results in small scale decentralised, partially observable settings. However results are not consistently *significantly* better than the baselines across tasks and settings. There are claims of superior performance in some tasks in the text but the the baselines have similar performance to the proposed method within standard error. For example, Figure 3 is selective to show results against SPARSE while CURIO performs similarly to ALIGN within error. Table 1 3v1 w/ keeper and Table 2 keep-away - CURIO baseline has similar performance within error. Symmetry-breaking performance is also not significant when scaled (appendix). In general the authors should be careful not to overstate claims unless there is statistical significance to back it up.
    - To improve the clarity of results I would suggest incorporating some learning dynamics curves. Only converged test rewards and task-specific converged metrics are currently given. Does coordination help with sample efficiency of training?
    - Comparison to Ndousse et al 2021 would also be useful to see whether this reward framing has a benefit over their proposed approach.
    - On line 161 there is a claim that the proxy dynamics model is validated in heterogeneous tasks but in the experiments it is shown that this is not the case in the experiments.
- *Clarity*:
    - The paper is well written in many parts. Paragraph headings and subsections are used effectively and the paper on the whole reads quite intuitively. Figure 1 is clear and informative to the reader.
    - To improve clarity:
        - the authors mention a gaussian noise experiment in the main text and refers to appendix but I cannot find this section in the results.
        - Figure 1: I would suggest increasing the font width for readability (particularly the equation)
        - The tables and figures are quite far away from their relevant sections in the text which hinders readability.
        - Line 293-294 : “analyse this in future experiments” - it is unclear if this means later in the paper or in future work.
    - A careful proofread is also required. Some issues I spotted:
        - Line 256: “a” not “an” goal state
        - Table 2 caption: misspelled “trained”
        - Iqbal & Sha 2019a and 2019b seem to refer to the same paper.
        - Line 160 misspelled “empirically”
        - Line 178: misspelled “team”
        - State and action space paragraph starting on Line 194 has a number of grammatical errors.
- *Significance:*
    - The paper addresses the problem of decentralised training which is less explored in the literature compared to centralised training but important for the scalability of MARL.
    - The alignment idea to use predictability of actions as an intrinsic reward is novel to my knowledge and well motivated. However, as the authors note in the related work section, there is a body of similar work that utilise model-based predictability in different ways e.g. as an auxiliary loss. It is not clear to me yet why using this inductive bias as a reward instead is significant, particularly without an experimental comparison.

---

> ### Author Response · Authors · 2022-08-02
> **Thank you for your positive, detailed comments and suggestions.**
>
> We address your questions below, and have updated the paper to reflect the recommended changes.
>
> > Did you compare this intrinsic reward based approach to other (non intrinsic reward based) approaches to the sparse learning problem (e.g. Ndousse et al 2021)? Some discussion around why the framing as an auxiliary reward instead of an auxiliary loss would be helpful.
>
> As you have already noted, our primary contribution is a multi-agent intrinsic reward. So, we standardized our training algorithm and model architectures to isolate the effect of different intrinsic rewards. Methods, such as Ndousse et al 2021 do not present an intrinsic reward but a new learning algorithm that modifies the underlying learning algorithm by adding an auxiliary loss. As such, our intrinsic reward can be added to Ndousse et al.’s method as additional training signal. Similarly, it can be added to non-actor-critic methods such as COMA (Foerster et al. 2018) and VDN (Sunehag 2017). However, this is out of scope for a single paper since it adds another dimensionality of exploration that is orthogonal to our paper’s contribution. We have briefly discussed such future directions of exploration in Section 5.4.
>
> > there is a body of similar work that utilise model-based predictability in different ways e.g. as an auxiliary loss. It is not clear to me yet why using this inductive bias as a reward instead is significant
>
> There are multiple dimensions along which multi-agent team performance can be improved. One of those directions is develop better reinforcement learning algorithms; another is to develop better intrinsic rewards for multi-agent teams. Our work is situated along the second direction. The model-based work falls along the first dimension. In the long run, it’s not clear which of these two directions or whether a combination of the two will be fruitful.
>
> > why does modelling from the other agent’s point of view, instead of directly predicting your own future state (which incorporates the other agents action) more helpful
>
> Thanks for the question. We included the following discussion in the paper:
>
> We hypothesize that ALIGN_team (modeling from other agents’ point of view) enables better multiagent exploration. ALIGN_self (predicting your own future states) simply incentivizes agents to visit familiar states they can predict well, which may not be useful in the multiagent scenario.
>
> Our empirical results support this hypothesis. Consider the tasks Physical deception and
> Cooperative navigation. Physical deception benefits more from exploration than cooperative navigation by design: In physical deception, agents are rewarded for exploring all the landmarks in order to prevent adversaries from knowing the goal. By contrast, in cooperative navigation, agents are penalized for exploring; agents should opt for the greedy action of navigating to the first empty goal they discover so the team can simultaneously occupy as many goals as possible. Indeed, we see ALIGN_team consistently outperforms ALIGN_self in Physical deception, while ALIGN_self beats ALIGN_team in Cooperative navigation (Table 1 and 2).
>
> > “ALIGN_team and ALIGN_self seem to work better than ALIGN_adv on some competitive tasks indicating that it is sometimes better to be predictable to an adversary which seems counterintuitive.”
>
> Not quite; in competitive settings, ALIGN_team/self always incentivizes agents to be predictable to its teammates/itself and not the adversaries. We will make this more clear in the updated paper version.
>
> > In the first paragraph of Section 4 there is discussion about gaining information outside of the agent’s receptive field through encountering surprising actions from other agents. How does the alignment reward help convey this information? Does this help at execution time?
>
> The alignment reward does not change the agent observability. During training, it does reward states where agents align predictions of their teammates and thus learn about how to infer information through their teammates’ behaviors. This results in a better policy to run at execution.
>
> > In the centralised regime experiments, did you use the actual dynamics model learnt by each agent to calculate the intrinsic reward or did you still use the individual agents dynamics model as a proxy as in the decentralised case?
>
> In the centralized experiments, we used the agent’s own dynamics model as a proxy. We decided on using the proxy to compare fairly to the decentralized method.
>
> To better understand whether the proxy model is reasonable to use, we ran an additional experiment using the true dynamics model. We found that both fared comparably to each other (ALIGNteam with the ground truth dynamics model gains 147.83 ± 14.27 test episode reward, whereas ALIGNteam dynamics model proxy achieves 141.04 ± 8.04 in Cooperative navigation (3v0)). This suggests that using the agent’s own dynamics models is a reasonable proxy.

---

> > ### Author Response · Authors · 2022-08-02
> > **continued...**
> >
> > > Do you have any suggestions for how to improve performance using alignment for heterogenous agents?
> >
> > This is still an open research question. One simple extension of our method could be to bolster the dynamics model by providing it with agent-specific parameters when making predictions. For example, we could learn specific parameters for agents with different capabilities. This parameter makes the assumption that we know the capabilities of different agents (their speeds, their sizes, etc.). A more realistic further extension of this idea could enable agents to predict the which parameters to choose when observing a neighboring agent and condition its dynamics model with the appropriate parameters to predict future states.
> >
> > Because this is out of scope of our current work, we leave this as an interesting direction for future work.
> >
> > > Did you evaluate zero shot coordination with CURIO models as well?
> >
> > Yes, we evaluated the CURIO baselines’ zero-shot performance. We’ve updated Figure 6 to include their performance.
> >
> > Minor changes:
> > - We have refined the claim in line 161 to state that “we validate its applicability in small-scale heterogeneous multi-agent tasks where agents have variable capabilities, although the methods perform similarly when more heterogeneous agents are added.” We also highlight a hypothesis for why the degradation happens in line 289 - 293.
> > - We have added new references in line 45 and 33 as suggested.
> > - We have added the details and results of the Gaussian noise experiment in the appendix. Gaussian noise is added to the dynamics model’s outputs to manipulate the accuracy of the dynamics model and measure how it degrades the performance of multi-agent teams using alignment.
> > - We have corrected the typos mentioned.

---

> > > ### Comment · Reviewer_oB8u · 2022-08-04
> > > **Thank you to the authors for your response**
> > >
> > > Thank you for your replies to my questions and updates to the manuscript. My main concerns have been addressed and clarified. I have updated my score accordingly.

---

### Official Review · Reviewer_whMR · 2022-07-11

**Rating:** 4
**Confidence:** 4
**Soundness:** 2 fair
**Presentation:** 3 good
**Contribution:** 2 fair

**Summary:**

The authors proposed an intrinsic reward that learns each agent to align with the predictions of neighbors in a fully decentralized way for improving multi-agent coordination. They compared the proposed method with the intrinsic reward-based baselines, e.g. curiosity-based approach in several environments including Google Research football, and provides some investigations to show the performance improvement of the proposed method.

**Questions:**

1. The proposed intrinsic reward is based on how much the neighbors predict the agent well. For the fully decentralized training, the authors replace the neighbors’ dynamic models with their own dynamic models. Eventually, the proposed intrinsic reward is the opposite of the novelty-based intrinsic reward approach, and thus the multiple agents will be trained to visit the familiar states that each agent can predict well. Although the approximation of the neighbors’ dynamic models with a proxy is empirically good, it is not exactly the same as the neighbors’ models. Each neighbor may predict the agent differently. Can we say that the proposed method enhances the alignment? the performance improvement can come from other reasons.

2. Since the proposed intrinsic reward is the opposite of the novelty-based intrinsic reward, can the proposed intrinsic reward degrade the exploration?

3. In general, the curiosity-based approaches are not good at the dense-reward environment. For a fair comparison, the authors should provide experiments on both dense/sparse environments.
4. Furthermore, it would be great if the authors provide the performances as the learning processes.
5. The authors should provide the experiments on the SMAC environment. Also, the intrinsic reward-based baselines such as [1] are missing. The authors should compare the proposed method with [1].
6. In Section 4.2, the authors restricted the future prediction from j’s point of view by using the portion of j’s observation i can see. The reviewer thinks that for the fully decentralized training, A should be unaware of how much B can observe. Is it okay that Agent i knows o_{i \and j} in the fully decentralized training?

[1] Zheng, Lulu, et al. "Episodic multi-agent reinforcement learning with curiosity-driven exploration." Advances in Neural Information Processing Systems 34 (2021): 3757-3769.

**Limitations:**

As mentioned in the “Questions”, the author should provide the experiments on the SMAC environment and compare the proposed method with the state-of-the-art baseline of the intrinsic reward-based MARL algorithm [1].

**Strengths And Weaknesses:**

Strengths

-	This paper is well written and easy to follow.
-	This paper deals with enhancing predictability of other agents in a fully decentralized way, which is a significant problem in MARL.
-	This paper provides some experiments to show why the proposed method works.

Weaknesses

-	The novelty of this paper is not enough for accepting the venue of NeurIPS.
-	There are some concerns of the proposed method (Please see “Questions”)
-	The lack of experiments on the popular benchmarks such as SMAC and the sparse environments.

---

> ### Author Response · Authors · 2022-08-02
> **We have added multiple new experiments and discussed your thoughtful questions in detail.**
>
> Thank you for your feedback and questions. We have revised the paper with (1) new experiments to validate the approximation of a neighbor’s dynamics model, (2) further discussion on exploration, (3) more experiments on dense environments, (4) added learning curves, (5) and responses to your remaining questions.
>
> > Can we say that the proposed method enhances the alignment? the performance improvement can come from other reasons.
>
> Yes, our experiments indicate that the performance improvements come from the proposed alignment reward. There are two experiments in particular that isolate this causal relationship:
>
> 1. The experiments across the 6 tasks in Table 1 and Table 2 compare sparse versus alignment. The only difference between these two approaches is the addition of alignment as an intrinsic reward, resulting in improvements in performance across all 6 tasks under partial observability and even when the number of agents are scaled. As reviewer oB8u also notes: “The same architectures and optimisation algorithms are chosen to provide a fair comparison against other intrinsic rewards.”
>
> 2. In addition, we have added a new experiment where we compare the ground truth alignment reward (using the true action of the other agents) against the proxy model (the current implementation presented in the paper ). This experiment shows that ALIGN_team with the ground truth dynamics model achieves a test episode reward of 147.83 ± 14.27 compared to the proxy model’s 141.04 ± 8.04 in Cooperative navigation (3v0), suggesting that our assumption of using an agent’s own dynamics model is a good proxy.
> We’ve updated the text in the paper to make this more clear.
>
> > Since the proposed intrinsic reward is the opposite of the novelty-based intrinsic reward, can the proposed intrinsic reward degrade the exploration?
>
> We have added the following discussion to the appendix:
>
> Although curiosity has proven useful for exploration in single-agent tasks, we find that alignment—which mathematically encourages agents to be more predictable instead of finding novelty—outperforms curiosity in numerous multi-agent tasks. We hypothesize that our results arise because today’s multi-agent task state space requires significantly less exploration than those used for single-agent (e.g. Atari games). Future work should develop new multi-agent environments that demand exploration complexity and where both curiosity and alignment would be necessary for collaboration. For example, in a search and rescue task where a single agent is unable to carry the injured, curiosity would encourage “search” while alignment would speed up “rescue”. In the end, we envision that both these forms of rewards would be necessary for successful collaboration. However, choosing when to encourage curiosity versus alignment is an open research problem.
>
> > In general, the curiosity-based approaches are not good at the dense-reward environment. For a fair comparison, the authors should provide experiments on both dense/sparse environments.
>
> Thanks for the suggestion! We’ve updated Table 1 to include model performance in the dense reward setting. We used the dense rewards from the original multi-particle environment (Lowe et al. 2017).
>
> Interestingly, despite hyperparameter tuning, we find that hand-crafted dense rewards perform better than alignment on only 3 of the 5 multi-particle environments. In both cooperative tasks, they perform even worse than sparse. We hypothesize this is due to the fact that the original dense rewards were tuned for MADDPG, the original learning algorithm used in Lowe et al. As we are using a multi-agent soft-actor critic, the hand-crafted reward values for the intermediate states would have to be re-tuned. To avoid falsely showcasing results when combining uncalibrated dense rewards with intrinsic rewards, we omit such experiments. We followed precedence set by prior work (Iqbal et al 2021) and only add intrinsic rewards to sparse rewards. This is a harder, ecologically valid scenario since most real world applications do not have hand crafted dense rewards available.
>
> > Furthermore, it would be great if the authors provide the performances as the learning processes.
>
> Thanks for the suggestion. We’ve added learning curves into the Appendix. Compared to the best baseline’s highest performance at the 100th epoch, we find that it only takes the best alignment variant 31 (Coop nav), 74 (Hetero nav), 30 (Physical dec), 97 (Predator-prey) and 92 (Keep-away) epochs respectively in the five multi-agent particle tasks. This means that on average the best alignment variant requires 35%±32 fewer training steps to reach the same performance as curiosity or sparse methods.

---

> > ### Author Response · Authors · 2022-08-02
> > **continued**
> >
> > > The authors should provide the experiments on the SMAC environment.
> >
> > Thank you for the suggestion. To directly compare against prior work on multiagent intrinsic rewards, we decided to focus the experiments on the environments that our baselines use. We additionally ran on the complex Google football environment, which we believe is equally complex to SMAC. We have already compared our method across 6 different tasks with varying complexity.
> >
> > > The authors should compare the proposed method with [1].
> >
> > Thank you for the suggestion. As reviewer oB8u also noted, our primary contribution is a novel multi-agent intrinsic reward function for decentralized training. [1] introduces a new learning algorithm, and not an intrinsic reward. Also, [1] requires centralized training. Adding a new dimension where we compare different intrinsic rewards against different reinforcement learning algorithms is out of scope for a single paper contribution.
> >
> >
> > > In Section 4.2, the authors restricted the future prediction from j’s point of view by using the portion of j’s observation i can see. The reviewer thinks that for the fully decentralized training, A should be unaware of how much B can observe. Is it okay that Agent i knows o_{i \and j} in the fully decentralized training?
> >
> > Thanks for raising an interesting point. Our work assumes that agents are aware of each other’s field of view range. During decentralized training, agent $i$ only needs its own observation (which includes agent $j$’s position if j is in i’s field of view) and agent $j$’s observable radius (same as its own) to infer the parts of its observation that j can observe. In many multi-agent tasks, this assumption will hold. For example, all players in a football team have roughly the same observability.
> >
> > However, we agree that as the number of autonomous agent hardware setups increases in the future, we will need to develop mechanisms to ascertain other’s field of view. We call out this assumption more explicitly in section 4.3.

---

> > > ### Comment · Reviewer_whMR · 2022-08-04
> > > **Response to the authors**
> > >
> > > Thanks for your feedback. Some concerns are solved, but some questions still remain.
> > >
> > >
> > > 1. From the results that alignment performs better than curiosity, it seems that reducing the uncertainty when the neighbors exist near is more important than exploration in the considered environments. However, I'm still concerned that the proposed method can perform worse in environments that need exploration, so it makes me hesitate to increase the score.
> > >
> > > 2-1. Is the sigma of policy (if the authors use Gaussian policy) trained to decrease faster if we use the proposed intrinsic reward?
> > > I think alignment makes the policy to be deterministic faster and thus it can perform better than naive MA-SAC. In this sense, did the authors tune for the temperature parameter ($\alpha$) in SAC? I'm concerned if the performance improvement may be made because alignment is equivalent to the effect of reducing $\alpha$.
> > >
> > > 2-2. In this sense, MASAC with $\alpha=0.1$ versus MASAC+Alignment with $\alpha=0.1$ is unfair.
> > > The authors should do hyperparameter tunings of MASAC, MASAC+alignment, and MASAC+curiosity separately.
> > >
> > >
> > > 3. Can the authors provide the performance of the SOTA algorithm in the Google football environment? The received score seems low.

---

> > > > ### Author Response · Authors · 2022-08-09
> > > > **Thank you for your follow-up questions. We appreciate your due diligence in this reviewing process.**
> > > >
> > > > > From the results that alignment performs better than curiosity, it seems that reducing the uncertainty when the neighbors exist near is more important than exploration in the considered environments. However, I'm still concerned that the proposed method can perform worse in environments that need exploration, so it makes me hesitate to increase the score.
> > > >
> > > > Our contributions should be held to the merits of our contribution, not the limitations of the existing benchmarks developed by others in the multi-agent community. Research progress relies on contributions that improve performance on existing benchmarks and on those that identify possible limitations in them. Our paper contributes both: we show improvements on 6 standard multi-agent benchmark tasks and also identify a limitation in those benchmarks—that newer multi-agent benchmarks should prioritize exploration. We hope that our findings will propel the development of future multi-agent benchmarks that necessitate exploration along with coordination.
> > > >
> > > > > did the authors tune for the temperature parameter (α) in SAC?
> > > >
> > > > We tuned the entropy regularization/temperature parameter (α) parameter for each algorithm. For MA-SAC and MA-SAC + alignment, we found α=0.1 leads to the best results. Meanwhile, MA-SAC+curiosity achieves its best result with α=0.05. In the Cooperative Navigation (3v0) task, the best ALIGN (α=0.1) scores 155.88±5.11, whereas SPARSE and the best CURIO (α=0.05) only score 114.93±26.31 and 137.29±6.80 respectively. These results suggest that alignment is not equivalent to the effect of reducing α.
> > > >
> > > > > performance of the SOTA algorithm in the Google football environment
> > > >
> > > > Our multi-agent performance is comparable to that of the SOTA algorithm in the Google football environment. Table 2 in [1] suggests that MADDPG (centralized) alone achieves an average return of 0.004±0.002 in the 3vs1-with-keeper task when trained with the “scoring” reward for 8M timesteps, while our decentralized MA-SAC (decentralized) alone achieves  0.020±0.001 when trained for 5M timesteps. Additionally, Figure 5(b) in [2] shows that IDQN’s (Tampuu et al. 2017) mean reward in the same task is around 0.05. Taken together, our reward scores are comparable to those in the literature. We get higher scores than (centralized) MADDPG with few training steps even though we use decentralized MA-SAC. However, IDQN is a much stronger learning algorithm than MASAC, leading to slightly better scores.

---

> > > > > ### Comment · Reviewer_whMR · 2022-08-10
> > > > > **Response to the authors**
> > > > >
> > > > > Thank you for the authors' response. Although I still think the authors should tune the temperature parameter in all considered environments,  I will increase my score.

---

### Official Review · Reviewer_D8qp · 2022-07-12

**Rating:** 7
**Confidence:** 4
**Soundness:** 4 excellent
**Presentation:** 4 excellent
**Contribution:** 3 good

**Summary:**

This paper tackles the problem of coordination in multi agent reinforcement learning, particularly in the decentralized training / partial observability setup. The paper proposes to derive a self-supervised reward from alignment. Agents are encouraged to take actions that match other agents' expectations when they cooperate and they are encouraged to do the opposite with competing agents.

For each agent the reward is computed using a dynamics model. This model is trained by predicting the next observation given the current action and observation. This model is then used to derive a reward that estimates the neighbors expectations of the agent's next state. The reward is decentralized and does not require the centralized training / decentralized execution which is common in the multi agent reinforcement learning literature.

The proposed reward is evaluated across many problems using the multi agent particules environment and Google Research football. The authors compare their algorithm with several existing baselines and the alignment based reward is shown to perform better than existing techniques for cooperative and competitive problems.


**Questions:**

L282 "When the number of agents increases, alignment scales well in all multi-agent particle tasks except for heterogenous navigation."
Do you expect this statement to remain true when the number of agents increases to hundreds or even thousands? In that case I would imagine that the summation L159 would make the alignment reward pretty noisy and coordination between agents would be more difficult.

**Limitations:**

The authors have adequately addressed the limitations and potential negative societal impact of their work.

**Strengths And Weaknesses:**

I found this paper to well be written and easy to follow. The authors made an effort to provide many helpful illustrative examples of the behaviour of the proposed technique. The method is simple and appears easy to implement.

The empirical evaluation of the paper is solid, it includes cooperative and competitive problems, with decentralized and centralized training and the proposed method is compared with several existing baselines. I found section 5.7 to be useful to understand how alignment reward helps and what are the limitations, particularly that the overall performance of the system decreases when the accuracy of the dynamics model decreases or that the method is less is competitive with heterogenous agents are used.

---

> ### Author Response · Authors · 2022-08-02
> **Thank you for your positive review, feedback, and questions.**
>
> > L282 "When the number of agents increases, alignment scales well in all multi-agent particle tasks except for heterogenous navigation." Do you expect this statement to remain true when the number of agents increases to hundreds or even thousands?
>
> While we empirically found that alignment scales well even when we increase the number of agents to 10, we have not yet tested our method in environments with hundreds or thousands of agents. We agree with reviewer D8qp that aligning to hundreds of agent’s expectations might be akin to aligning to noise. In Zoology (Couzin, 2007), the commonly accepted model for animal collective behavior assumes that animals only align to their N closest neighbors, where N is much smaller than a hundred. Our alignment implementation operationalizes this finding by rewarding aligning behavior with teammates within an agent’s field of view. However, we have yet to test the dynamics of alignment in massive collaboration settings but have added this suggestion to the discussion section.

---

> > ### Comment · Reviewer_D8qp · 2022-08-09
> > **Response to authors**
> >
> > Thank you for response. After reading other reviews and the rebuttal I will keep my score as is.

---

### Meta-Review · Area_Chair_G3Ec · 2022-08-24

**Recommendation:** Accept
**Confidence:** Certain

**Metareview:**

This paper introduces a novel method for decentralised training in both cooperative and competitive environments.
The main insight is that agents should be predictable to their team mates but unpredictable to adversaries.
Crucially, each agent relies on a local model to optimise this objective, making it compatible with decentralised training.

There were a few concerns from reviewers around both how the method applies to the decentralised training regime and regarding the naming. The authors managed to address the concerns appropriately, leading the actively engaged reviewers to increase their scores substantially. Reviewer TEM3 kept their rating at a borderline reject even though the concerns were addressed by the rebuttal. As such I recommend to discard this review. Reviewer whMR stated that they would increase their score but to the best of my knowledge didn't do so. Again, their review should be seen in this context.


**Award:**

No

---

### Decision · Program_Chairs · 2022-09-14

Accept